# Multilabel Structured Output Learning with Random Spanning Trees of Max-Margin Markov Networks

**Mario Marchand**
Département d'informatique et génie logiciel
Université Laval
Québec (QC), Canada
mario.marchand@ift.ulaval.ca

**Hongyu Su**
Helsinki Institute for Information Technology
Dept of Information and Computer Science
Aalto University, Finland
hongyu.su@aalto.fi

**Emilie Morvant**[*]
LaHC, UMR CNRS 5516
Univ. of St-Etienne, France
emilie.morvant@univ-st-etienne.fr

**Juho Rousu**
Helsinki Institute for Information Technology
Dept of Information and Computer Science
Aalto University, Finland
juho.rousu@aalto.fi

**John Shawe-Taylor**
Department of Computer Science
University College London
London, UK
j.shawe-taylor@ucl.ac.uk

## Abstract

We show that the usual score function for conditional Markov networks can be written as the expectation over the scores of their spanning trees. We also show that a small random sample of these output trees can attain a significant fraction of the margin obtained by the complete graph and we provide conditions under which we can perform tractable inference. The experimental results confirm that practical learning is scalable to realistic datasets using this approach.

## 1 Introduction

Finding an hyperplane that minimizes the number of misclassifications is $\mathcal{NP}$-hard. But the support vector machine (SVM) substitutes the hinge for the discrete loss and, modulo a margin assumption, can nonetheless efficiently find a hyperplane with a guarantee of good generalization. This paper investigates whether the problem of inference over a complete graph in structured output prediction can be avoided in an analogous way based on a margin assumption.

We first show that the score function for the complete output graph can be expressed as the expectation over the scores of random spanning trees. A sampling result then shows that a small random sample of these output trees can attain a significant fraction of the margin obtained by the complete graph. Together with a generalization bound for the sample of trees, this shows that we can obtain good generalization using the average scores of a sample of trees in place of the complete graph. We have thus reduced the intractable inference problem to a convex optimization not dissimilar to a SVM. The key inference problem to enable learning with this ensemble now becomes finding the maximum violator for the (finite sample) average tree score. We then provide the conditions under which the inference problem is tractable. Experimental results confirm this prediction and show that

---

[*]Most of this work was carried out while E. Morvant was affiliated with IST Austria, Klosterneurburg.

practical learning is scalable to realistic datasets using this approach with the resulting classification accuracy enhanced over more naive ways of training the individual tree score functions.

The paper aims at exploring the potential ramifications of the random spanning tree observation both theoretically and practically. As such, we think that we have laid the foundations for a fruitful approach to tackle the intractability of inference in a number of scenarios. Other attractive features are that we do not require knowledge of the output graph's structure, that the optimization is convex, and that the accuracy of the optimization can be traded against computation. Our approach is firmly rooted in the maximum margin Markov network analysis [1]. Other ways to address the intractability of loopy graph inference have included using approximate MAP inference with tree-based and LP relaxations [2], semi-definite programming convex relaxations [3], special cases of graph classes for which inference is efficient [4], use of random tree score functions in heuristic combinations [5]. Our work is not based on any of these approaches, despite superficial resemblances to, *e.g.,* the trees in tree-based relaxations and the use of random trees in [5]. We believe it represents a distinct approach to a fundamental problem of learning and, as such, is worthy of further investigation.

## 2  Definitions and Assumptions

We consider supervised learning problems where the input space $\mathcal{X}$ is arbitrary and the output space $\mathcal{Y}$ consists of the set of all $\ell$-dimensional multilabel vectors $(y_1, \ldots, y_\ell) \stackrel{\text{def}}{=} \mathbf{y}$ where each $y_i \in \{1, \ldots, r_i\}$ for some finite positive integer $r_i$. Each example $(x, \mathbf{y}) \in \mathcal{X} \times \mathcal{Y}$ is mapped to a joint feature vector $\boldsymbol{\phi}(x, \mathbf{y})$. Given a weight vector $\mathbf{w}$ in the space of joint feature vectors, the predicted output $\mathbf{y_w}(x)$ at input $x \in \mathcal{X}$, is given by the output $\mathbf{y}$ maximizing the *score* $F(\mathbf{w}, x, \mathbf{y})$, *i.e.*,

$$\mathbf{y_w}(x) \stackrel{\text{def}}{=} \operatorname*{argmax}_{\mathbf{y} \in \mathcal{Y}} F(\mathbf{w}, x, \mathbf{y}) \quad ; \quad \text{where} \quad F(\mathbf{w}, x, \mathbf{y}) \stackrel{\text{def}}{=} \langle \mathbf{w}, \boldsymbol{\phi}(x, \mathbf{y}) \rangle , \tag{1}$$

and where $\langle \cdot, \cdot \rangle$ denotes the inner product in the joint feature space. Hence, $\mathbf{y_w}(x)$ is obtained by solving the so-called *inference* problem, which is known to be $\mathcal{NP}$-hard for many output feature maps [6, 7]. Consequently, we aim at using an output feature map for which the inference problem can be solved by a polynomial time algorithm such as dynamic programming. The *margin* $\Gamma(\mathbf{w}, x, \mathbf{y})$ achieved by predictor $\mathbf{w}$ at example $(x, \mathbf{y})$ is defined as,

$$\Gamma(\mathbf{w}, x, \mathbf{y}) \quad \stackrel{\text{def}}{=} \quad \min_{\mathbf{y}' \neq \mathbf{y}} \left[ F(\mathbf{w}, x, \mathbf{y}) - F(\mathbf{w}, x, \mathbf{y}') \right] .$$

We consider the case where the feature map $\boldsymbol{\phi}$ is a potential function for a Markov network defined by a complete graph $G$ with $\ell$ nodes and $\ell(\ell - 1)/2$ undirected edges. Each node $i$ of $G$ represents an output variable $y_i$ and there exists an edge $(i, j)$ of $G$ for each pair $(y_i, y_j)$ of output variables. For any example $(x, \mathbf{y}) \in \mathcal{X} \times \mathcal{Y}$, its joint feature vector is given by

$$\boldsymbol{\phi}(x, \mathbf{y}) = \big( \boldsymbol{\phi}_{i,j}(x, y_i, y_j) \big)_{(i,j) \in G} = \big( \boldsymbol{\varphi}(x) \otimes \boldsymbol{\psi}_{i,j}(y_i, y_j) \big)_{(i,j) \in G} ,$$

where $\otimes$ is the Kronecker product. Hence, any predictor $\mathbf{w}$ can be written as $\mathbf{w} = (\mathbf{w}_{i,j})_{(i,j) \in G}$ where $\mathbf{w}_{i,j}$ is $\mathbf{w}$'s weight on $\boldsymbol{\phi}_{i,j}(x, y_i, y_j)$. Therefore, for any $\mathbf{w}$ and any $(x, \mathbf{y})$, we have

$$F(\mathbf{w}, x, \mathbf{y}) = \langle \mathbf{w}, \boldsymbol{\phi}(x, \mathbf{y}) \rangle = \sum_{(i,j) \in G} \langle \mathbf{w}_{i,j}, \boldsymbol{\phi}_{i,j}(x, y_i, y_j) \rangle = \sum_{(i,j) \in G} F_{i,j}(\mathbf{w}_{i,j}, x, y_i, y_j) ,$$

where we denote by $F_{i,j}(\mathbf{w}_{i,j}, x, y_i, y_j) = \langle \mathbf{w}_{i,j}, \boldsymbol{\phi}_{i,j}(x, y_i, y_j) \rangle$ the score of labeling the edge $(i, j)$ by $(y_i, y_j)$ given input $x$.

For any vector $\mathbf{a}$, let $\|\mathbf{a}\|$ denote its $L_2$ norm. Throughout the paper, we make the assumption that we have a normalized joint feature space such that $\|\boldsymbol{\phi}(x, \mathbf{y})\| = 1$ for all $(x, \mathbf{y}) \in \mathcal{X} \times \mathcal{Y}$ and $\|\boldsymbol{\phi}_{i,j}(x, y_i, y_j)\|$ is the same for all $(i, j) \in G$. Since the complete graph $G$ has $\binom{\ell}{2}$ edges, it follows that $\|\boldsymbol{\phi}_{i,j}(x, y_i, y_j)\|^2 = \binom{\ell}{2}^{-1}$ for all $(i, j) \in G$.

We also have a training set $S \stackrel{\text{def}}{=} \{(x_1, \mathbf{y}_1), \ldots, (x_m, \mathbf{y}_m)\}$ where each example is generated independently according to some unknown distribution $D$. Mathematically, we do not assume the existence of a predictor $\mathbf{w}$ achieving some positive margin $\Gamma(\mathbf{w}, x, \mathbf{y})$ on each $(x, \mathbf{y}) \in S$. Indeed,

for some $S$, there might not exist any $\mathbf{w}$ where $\Gamma(\mathbf{w}, x, \mathbf{y}) > 0$ for all $(x, \mathbf{y}) \in S$. However, the generalization guarantee will be best when $\mathbf{w}$ achieves a large margin on most training points.

Given any $\gamma > 0$, and any $(x, \mathbf{y}) \in \mathcal{X} \times \mathcal{Y}$, the *hinge loss* (at scale $\gamma$) incurred on $(x, \mathbf{y})$ by a unit $L_2$ norm predictor $\mathbf{w}$ that achieves a (possibly negative) margin $\Gamma(\mathbf{w}, x, \mathbf{y})$ is given by $\mathcal{L}^\gamma(\Gamma(\mathbf{w}, x, \mathbf{y}))$, where the so-called *hinge loss function* $\mathcal{L}^\gamma$ is defined as $\mathcal{L}^\gamma(s) \overset{\text{def}}{=} \max(0, 1 - s/\gamma) \; \forall s \in \mathbb{R}$. We will also make use of the *ramp loss function* $\mathcal{A}^\gamma$ defined by $\mathcal{A}^\gamma(s) \overset{\text{def}}{=} \min(1, \mathcal{L}^\gamma(s)) \; \forall s \in \mathbb{R}$.

*The proofs of all the rigorous results of this paper are provided in the supplementary material.*

## 3 Superposition of Random Spanning Trees

Given a complete graph $G$ of $\ell$ nodes (representing the Markov network), let $S(G)$ denote the set of all $\ell^{\ell-2}$ spanning trees of $G$. Recall that each spanning tree of $G$ has $\ell - 1$ edges. Hence, for any edge $(i, j) \in G$, the number of trees in $S(G)$ covering that edge $(i, j)$ is given by $\ell^{\ell-2}(\ell-1)/\binom{\ell}{2} = (2/\ell)\ell^{\ell-2}$. Therefore, for any function $f$ of the edges of $G$ we have

$$\sum_{T \in S(G)} \sum_{(i,j) \in T} f((i,j)) = \ell^{\ell-2} \frac{2}{\ell} \sum_{(i,j) \in G} f((i,j)) \ .$$

Given any spanning tree $T$ of $G$ and given any predictor $\mathbf{w}$, let $\mathbf{w}_T$ denote the projection of $\mathbf{w}$ on the edges of $T$. Namely, $(\mathbf{w}_T)_{i,j} = \mathbf{w}_{i,j}$ if $(i, j) \in T$, and $(\mathbf{w}_T)_{i,j} = 0$ otherwise. Let us also denote by $\boldsymbol{\phi}_T(x, \mathbf{y})$, the projection of $\boldsymbol{\phi}(x, \mathbf{y})$ on the edges of $T$. Namely, $(\boldsymbol{\phi}_T(x, \mathbf{y}))_{i,j} = \boldsymbol{\phi}_{i,j}(x, y_i, y_j)$ if $(i, j) \in T$, and $(\boldsymbol{\phi}_T(x, \mathbf{y}))_{i,j} = 0$ otherwise. Recall that $\|\boldsymbol{\phi}_{i,j}(x, y_i, y_j)\|^2 = \binom{\ell}{2}^{-1} \; \forall (i, j) \in G$. Thus, for all $(x, \mathbf{y}) \in \mathcal{X} \times \mathcal{Y}$ and for all $T \in S(G)$, we have

$$\|\boldsymbol{\phi}_T(x, \mathbf{y})\|^2 = \sum_{(i,j) \in T} \|\boldsymbol{\phi}_{i,j}(x, y_i, y_j)\|^2 = \frac{\ell - 1}{\binom{\ell}{2}} = \frac{2}{\ell} \ .$$

We now establish how $F(\mathbf{w}, x, \mathbf{y})$ can be written as an expectation over all the spanning trees of $G$.

**Lemma 1.** *Let $\hat{\mathbf{w}}_T \overset{\text{def}}{=} \mathbf{w}_T/\|\mathbf{w}_T\|$, $\hat{\boldsymbol{\phi}}_T \overset{\text{def}}{=} \boldsymbol{\phi}_T/\|\boldsymbol{\phi}_T\|$. Let $\mathcal{U}(G)$ denote the uniform distribution on $S(G)$. Then, we have*

$$F(\mathbf{w}, x, \mathbf{y}) = \operatorname*{\mathbf{E}}_{T \sim \mathcal{U}(G)} a_T \langle \hat{\mathbf{w}}_T, \hat{\boldsymbol{\phi}}_T(x, \mathbf{y}) \rangle, \quad \text{where } a_T \overset{\text{def}}{=} \sqrt{\frac{\ell}{2}} \, \|\mathbf{w}_T\| \ .$$

*Moreover, for any $\mathbf{w}$ such that $\|\mathbf{w}\| = 1$, we have:* $\operatorname*{\mathbf{E}}_{T \sim \mathcal{U}(G)} a_T^2 = 1$, *and* $\operatorname*{\mathbf{E}}_{T \sim \mathcal{U}(G)} a_T \leq 1$.

Let $\mathcal{T} \overset{\text{def}}{=} \{T_1, \dots, T_n\}$ be a sample of $n$ spanning trees of $G$ where each $T_i$ is sampled independently according to $\mathcal{U}(G)$. Given any unit $L_2$ norm predictor $\mathbf{w}$ on the complete graph $G$, our task is to investigate how the margins $\Gamma(\mathbf{w}, x, \mathbf{y})$, for each $(x, \mathbf{y}) \in \mathcal{X} \times \mathcal{Y}$, will be modified if we approximate the (true) expectation over all spanning trees by an average over the sample $\mathcal{T}$.

For this task, we consider any $(x, \mathbf{y})$ and any $\mathbf{w}$ of unit $L_2$ norm. Let $F_\mathcal{T}(\mathbf{w}, x, \mathbf{y})$ denote the estimation of $F(\mathbf{w}, x, \mathbf{y})$ on the tree sample $\mathcal{T}$,

$$F_\mathcal{T}(\mathbf{w}, x, \mathbf{y}) \overset{\text{def}}{=} \frac{1}{n} \sum_{i=1}^n a_{T_i} \langle \hat{\mathbf{w}}_{T_i}, \hat{\boldsymbol{\phi}}_{T_i}(x, \mathbf{y}) \rangle \ ,$$

and let $\Gamma_\mathcal{T}(\mathbf{w}, x, \mathbf{y})$ denote the estimation of $\Gamma(\mathbf{w}, x, \mathbf{y})$ on the tree sample $\mathcal{T}$,

$$\Gamma_\mathcal{T}(\mathbf{w}, x, \mathbf{y}) \overset{\text{def}}{=} \min_{\mathbf{y}' \neq \mathbf{y}} [F_\mathcal{T}(\mathbf{w}, x, \mathbf{y}) - F_\mathcal{T}(\mathbf{w}, x, \mathbf{y}')] \ .$$

The following lemma states how $\Gamma_\mathcal{T}$ relates to $\Gamma$.

**Lemma 2.** *Consider any unit $L_2$ norm predictor $\mathbf{w}$ on the complete graph $G$ that achieves a margin of $\Gamma(\mathbf{w}, x, \mathbf{y})$ for each $(x, \mathbf{y}) \in \mathcal{X} \times \mathcal{Y}$, then we have*

$$\Gamma_\mathcal{T}(\mathbf{w}, x, \mathbf{y}) \geq \Gamma(\mathbf{w}, x, \mathbf{y}) - 2\epsilon \quad \forall (x, \mathbf{y}) \in \mathcal{X} \times \mathcal{Y} \ ,$$

*whenever we have $|F_\mathcal{T}(\mathbf{w}, x, \mathbf{y}) - F(\mathbf{w}, x, \mathbf{y})| \leq \epsilon$ for all $(x, \mathbf{y}) \in \mathcal{X} \times \mathcal{Y}$.*

Lemma 2 has important consequences whenever $|F_{\mathcal{T}}(\mathbf{w}, x, \mathbf{y}) - F(\mathbf{w}, x, \mathbf{y})| \leq \epsilon$ for all $(x, \mathbf{y}) \in \mathcal{X} \times \mathcal{Y}$. Indeed, if $\mathbf{w}$ achieves a hard margin $\Gamma(\mathbf{w}, x, \mathbf{y}) \geq \gamma > 0$ for all $(x, \mathbf{y}) \in S$, then we have that $\mathbf{w}$ also achieves a hard margin of $\Gamma_{\mathcal{T}}(\mathbf{w}, x, \mathbf{y}) \geq \gamma - 2\epsilon$ on each $(x, \mathbf{y}) \in S$ when using the tree sample $\mathcal{T}$ instead of the full graph $G$. More generally, if $\mathbf{w}$ achieves a ramp loss of $\mathcal{A}^{\gamma}(\Gamma(\mathbf{w}, x, \mathbf{y}))$ for each $(x, \mathbf{y}) \in \mathcal{X} \times \mathcal{Y}$, then $\mathbf{w}$ achieves a ramp loss of $\mathcal{A}^{\gamma}(\Gamma_{\mathcal{T}}(\mathbf{w}, x, \mathbf{y})) \leq \mathcal{A}^{\gamma}(\Gamma(\mathbf{w}, x, \mathbf{y}) - 2\epsilon)$ for all $(x, \mathbf{y}) \in \mathcal{X} \times \mathcal{Y}$ when using the tree sample $\mathcal{T}$ instead of the full graph $G$. This last property follows directly from the fact that $\mathcal{A}^{\gamma}(s)$ is a non-increasing function of $s$.

The next lemma tells us that, apart from a slow $\ln^2(\sqrt{n})$ dependence, a sample of $n \in \Theta(\ell^2/\epsilon^2)$ spanning trees is sufficient to assure that the condition of Lemma 2 holds with high probability for all $(x, \mathbf{y}) \in \mathcal{X} \times \mathcal{Y}$. Such a fast convergence rate was made possible by using PAC-Bayesian methods which, in our case, prevented us of using the union bound over all possible $\mathbf{y} \in \mathcal{Y}$.

**Lemma 3.** *Consider any $\epsilon > 0$ and any unit $L_2$ norm predictor $\mathbf{w}$ for the complete graph $G$ acting on a normalized joint feature space. For any $\delta \in (0, 1)$, let*

$$ n \geq \frac{\ell^2}{\epsilon^2} \left( \frac{1}{16} + \frac{1}{2} \ln \frac{8\sqrt{n}}{\delta} \right)^2 . \tag{2} $$

*Then with probability of at least $1 - \delta/2$ over all samples $\mathcal{T}$ generated according to $\mathcal{U}(G)^n$, we have, simultaneously for all $(x, \mathbf{y}) \in \mathcal{X} \times \mathcal{Y}$, that $|F_{\mathcal{T}}(\mathbf{w}, x, \mathbf{y}) - F(\mathbf{w}, x, \mathbf{y})| \leq \epsilon$.*

Given a sample $\mathcal{T}$ of $n$ spanning trees of $G$, we now consider an arbitrary set $\mathcal{W} \overset{\text{def}}{=} \{\hat{\mathbf{w}}_{T_1}, \ldots, \hat{\mathbf{w}}_{T_n}\}$ of unit $L_2$ norm weight vectors where each $\hat{\mathbf{w}}_{T_i}$ operates on a unit $L_2$ norm feature vector $\hat{\boldsymbol{\phi}}_{T_i}(x, \mathbf{y})$. For any $\mathcal{T}$ and any such set $\mathcal{W}$, we consider an arbitrary unit $L_2$ norm conical combination of each weight in $\mathcal{W}$ realized by a $n$-dimensional weight vector $\mathbf{q} \overset{\text{def}}{=} (q_1, \ldots, q_n)$, where $\sum_{i=1}^{n} q_i^2 = 1$ and each $q_i \geq 0$. Given any $(x, \mathbf{y})$ and any $\mathcal{T}$, we define the score $F_{\mathcal{T}}(\mathcal{W}, \mathbf{q}, x, \mathbf{y})$ achieved on $(x, \mathbf{y})$ by the conical combination $(\mathcal{W}, \mathbf{q})$ on $\mathcal{T}$ as

$$ F_{\mathcal{T}}(\mathcal{W}, \mathbf{q}, x, \mathbf{y}) \overset{\text{def}}{=} \frac{1}{\sqrt{n}} \sum_{i=1}^{n} q_i \langle \hat{\mathbf{w}}_{T_i}, \hat{\boldsymbol{\phi}}_{T_i}(x, \mathbf{y}) \rangle , \tag{3} $$

where the $\sqrt{n}$ denominator ensures that we always have $F_{\mathcal{T}}(\mathcal{W}, \mathbf{q}, x, \mathbf{y}) \leq 1$ in view of the fact that $\sum_{i=1}^{n} q_i$ can be as large as $\sqrt{n}$. Note also that $F_{\mathcal{T}}(\mathcal{W}, \mathbf{q}, x, \mathbf{y})$ is the score of the feature vector obtained by the concatenation of all the weight vectors in $\mathcal{W}$ (and weighted by $\mathbf{q}$) acting on a feature vector obtained by concatenating each $\hat{\boldsymbol{\phi}}_{T_i}$ multiplied by $1/\sqrt{n}$. Hence, given $\mathcal{T}$, we define the margin $\Gamma_{\mathcal{T}}(\mathcal{W}, \mathbf{q}, x, \mathbf{y})$ achieved on $(x, \mathbf{y})$ by the conical combination $(\mathcal{W}, \mathbf{q})$ on $\mathcal{T}$ as

$$ \Gamma_{\mathcal{T}}(\mathcal{W}, \mathbf{q}, x, \mathbf{y}) \overset{\text{def}}{=} \min_{\mathbf{y}' \neq \mathbf{y}} \left[ F_{\mathcal{T}}(\mathcal{W}, \mathbf{q}, x, \mathbf{y}) - F_{\mathcal{T}}(\mathcal{W}, \mathbf{q}, x, \mathbf{y}') \right] . \tag{4} $$

For any unit $L_2$ norm predictor $\mathbf{w}$ that achieves a margin of $\Gamma(\mathbf{w}, x, \mathbf{y})$ for all $(x, \mathbf{y}) \in \mathcal{X} \times \mathcal{Y}$, we now show that there exists, with high probability, a unit $L_2$ norm conical combination $(\mathcal{W}, \mathbf{q})$ on $\mathcal{T}$ achieving margins that are not much smaller than $\Gamma(\mathbf{w}, x, \mathbf{y})$.

**Theorem 4.** *Consider any unit $L_2$ norm predictor $\mathbf{w}$ for the complete graph $G$, acting on a normalized joint feature space, achieving a margin of $\Gamma(\mathbf{w}, x, \mathbf{y})$ for each $(x, \mathbf{y}) \in \mathcal{X} \times \mathcal{Y}$. Then for any $\epsilon > 0$, and any $n$ satisfying Lemma 3, for any $\delta \in (0, 1]$, with probability of at least $1 - \delta$ over all samples $\mathcal{T}$ generated according to $\mathcal{U}(G)^n$, there exists a unit $L_2$ norm conical combination $(\mathcal{W}, \mathbf{q})$ on $\mathcal{T}$ such that, simultaneously for all $(x, \mathbf{y}) \in \mathcal{X} \times \mathcal{Y}$, we have*

$$ \Gamma_{\mathcal{T}}(\mathcal{W}, \mathbf{q}, x, \mathbf{y}) \geq \frac{1}{\sqrt{1 + \epsilon}} \left[ \Gamma(\mathbf{w}, x, \mathbf{y}) - 2\epsilon \right] . $$

From Theorem 4, and since $\mathcal{A}^{\gamma}(s)$ is a non-increasing function of $s$, it follows that, with probability at least $1 - \delta$ over the random draws of $\mathcal{T} \sim \mathcal{U}(G)^n$, there exists $(\mathcal{W}, \mathbf{q})$ on $\mathcal{T}$ such that, simultaneously for all $\forall (x, \mathbf{y}) \in \mathcal{X} \times \mathcal{Y}$, for any $n$ satisfying Lemma 3 we have

$$ \mathcal{A}^{\gamma}(\Gamma_{\mathcal{T}}(\mathcal{W}, \mathbf{q}, x, \mathbf{y})) \leq \mathcal{A}^{\gamma} \left( \left[ \Gamma(\mathbf{w}, x, \mathbf{y}) - 2\epsilon \right] (1 + \epsilon)^{-1/2} \right) . $$

Hence, instead of searching for a predictor $\mathbf{w}$ for the complete graph $G$ that achieves a small expected ramp loss $\mathbf{E}_{(x, \mathbf{y}) \sim D} \mathcal{A}^{\gamma}(\Gamma(\mathbf{w}, x, \mathbf{y}))$, Theorem 4 tells us that we can settle the search for a

unit $L_2$ norm conical combination $(\mathcal{W}, \mathbf{q})$ on a sample $\mathcal{T}$ of randomly-generated spanning trees of $G$ that achieves small $\mathbf{E}_{(x,\mathbf{y})\sim D}\mathcal{A}^\gamma(\Gamma_\mathcal{T}(\mathcal{W}, \mathbf{q}, x, \mathbf{y}))$. But recall that $\Gamma_\mathcal{T}(\mathcal{W}, \mathbf{q}, x, \mathbf{y}))$ is the margin of a weight vector obtained by the concatenation of all the weight vectors in $\mathcal{W}$ (weighted by $\mathbf{q}$) on a feature vector obtained by the concatenation of the $n$ feature vectors $(1/\sqrt{n})\hat{\boldsymbol{\phi}}_{T_i}$. It thus follows that any standard risk bound for the SVM applies directly to $\mathbf{E}_{(x,\mathbf{y})\sim D}\mathcal{A}^\gamma(\Gamma_\mathcal{T}(\mathcal{W}, \mathbf{q}, x, \mathbf{y}))$. Hence, by adapting the SVM risk bound of [8], we have the following result.

**Theorem 5.** *Consider any sample $\mathcal{T}$ of $n$ spanning trees of the complete graph $G$. For any $\gamma > 0$ and any $0 < \delta \leq 1$, with probability of at least $1 - \delta$ over the random draws of $S \sim D^m$, simultaneously for all unit $L_2$ norm conical combinations $(\mathcal{W}, \mathbf{q})$ on $\mathcal{T}$, we have*

$$\mathop{\mathbf{E}}_{(x,\mathbf{y})\sim D} \mathcal{A}^\gamma(\Gamma_\mathcal{T}(\mathcal{W}, \mathbf{q}, x, \mathbf{y})) \leq \frac{1}{m}\sum_{i=1}^m \mathcal{A}^\gamma(\Gamma_\mathcal{T}(\mathcal{W}, \mathbf{q}, x_i, \mathbf{y}_i)) + \frac{2}{\gamma\sqrt{m}} + 3\sqrt{\frac{\ln(2/\delta)}{2m}}\,.$$

Hence, according to this theorem, the conical combination $(\mathcal{W}, \mathbf{q})$ having the best generalization guarantee is the one which minimizes the sum of the first two terms on the right hand side of the inequality. Note that the theorem is still valid if we replace, in the empirical risk term, the non-convex ramp loss $\mathcal{A}^\gamma$ by the convex hinge loss $\mathcal{L}^\gamma$. This provides the theoretical basis of the proposed optimization problem for learning $(\mathcal{W}, \mathbf{q})$ on the sample $\mathcal{T}$.

## 4 A $L_2$-Norm Random Spanning Tree Approximation Approach

If we introduce the usual slack variables $\xi_k \stackrel{\text{def}}{=} \gamma \cdot \mathcal{L}^\gamma(\Gamma_\mathcal{T}(\mathcal{W}, \mathbf{q}, x_k, \mathbf{y}_k)$, Theorem 5 suggests that we should minimize $\frac{1}{\gamma}\sum_{k=1}^m \xi_k$ for some fixed margin value $\gamma > 0$. Rather than performing this task for several values of $\gamma$, we show in the supplementary material that we can, equivalently, solve the following optimization problem for several values of $C > 0$.

**Definition 6. Primal $L_2$-norm Random Tree Approximation.**

$$\begin{aligned} \mathop{\boldsymbol{min}}_{\mathbf{w}_{T_i},\xi_k} \quad & \frac{1}{2}\sum_{i=1}^n \|\mathbf{w}_{T_i}\|_2^2 + C\sum_{k=1}^m \xi_k \\ \boldsymbol{s.t.} \quad & \sum_{i=1}^n \langle \mathbf{w}_{T_i}, \hat{\boldsymbol{\phi}}_{T_i}(x_k, \mathbf{y}_k)\rangle - \mathop{\boldsymbol{max}}_{\mathbf{y}\neq\mathbf{y}_k}\sum_{i=1}^n \langle \mathbf{w}_{T_i}, \hat{\boldsymbol{\phi}}_{T_i}(x_k, \mathbf{y})\rangle \geq 1 - \xi_k, \\ & \xi_k \geq 0\,, \forall\, k \in \{1,\dots,m\}, \end{aligned}$$

*where $\{\mathbf{w}_{T_i}|T_i \in \mathcal{T}\}$ are the feature weights to be learned on each tree, $\xi_k$ is the margin slack allocated for each $x_k$, and $C$ is the slack parameter that controls the amount of regularization.*

This primal form has the interpretation of maximizing the joint margins from individual trees between (correct) training examples and all the other (incorrect) examples.

The key for the efficient optimization is solving the 'argmax' problem efficiently. In particular, we note that the space of all multilabels is exponential in size, thus forbidding exhaustive enumeration over it. In the following, we show how exact inference over a collection $\mathcal{T}$ of trees can be implemented in $\Theta(Kn\ell)$ time per data point, where $K$ is the smallest number such that the average score of the $K$'th best multilabel for each tree of $\mathcal{T}$ is at most $F_\mathcal{T}(x, \mathbf{y}) \stackrel{\text{def}}{=} \frac{1}{n}\sum_{i=1}^n \langle \mathbf{w}_{T_i}, \hat{\boldsymbol{\phi}}_{T_i}(x, \mathbf{y})\rangle$. Whenever $K$ is polynomial in the number of labels, this gives us exact polynomial-time inference over the ensemble of trees.

### 4.1 Fast inference over a collection of trees

It is well known that the exact solution to the inference problem

$$\hat{\mathbf{y}}_{T_i}(x) = \mathop{\text{argmax}}_{\mathbf{y}\in\mathcal{Y}} F_{\mathbf{w}_{T_i}}(x, \mathbf{y}) \stackrel{\text{def}}{=} \mathop{\text{argmax}}_{\mathbf{y}\in\mathcal{Y}} \langle \mathbf{w}_{T_i}, \hat{\boldsymbol{\phi}}_{T_i}(x, \mathbf{y})\rangle, \quad (5)$$

on an individual tree $T_i$ can be obtained in $\Theta(\ell)$ time by dynamic programming. However, there is no guarantee that the maximizer $\hat{\mathbf{y}}_{T_i}$ of Equation (5) is also a maximizer of $F_\mathcal{T}$. In practice, $\hat{\mathbf{y}}_{T_i}$

can differ for each spanning tree $T_i \in \mathcal{T}$. Hence, instead of using only the best scoring multilabel $\hat{\mathbf{y}}_{T_i}$ from each individual $T_i \in \mathcal{T}$, we consider the set of the $K$ highest scoring multilabels $\mathcal{Y}_{T_i,K} = \{\hat{\mathbf{y}}_{T_i,1}, \cdots, \hat{\mathbf{y}}_{T_i,K}\}$ of $F_{\mathbf{w}_{T_i}}(x,\mathbf{y})$. In the supplementary material we describe a dynamic programming to find the $K$ highest multilabels in $\Theta(K\ell)$ time. Running this algorithm for all of the trees gives us a candidate set of $\Theta(Kn)$ multilabels $\mathcal{Y}_{\mathcal{T},K} = \mathcal{Y}_{T_1,K} \cup \cdots \cup \mathcal{Y}_{T_n,K}$. We now state a key lemma that will enable us to verify if the candidate set contains the maximizer of $F_{\mathcal{T}}$.

**Lemma 7.** *Let* $\mathbf{y}_K^\star = \underset{\mathbf{y} \in \mathcal{Y}_{\mathcal{T},K}}{\operatorname{argmax}} F_{\mathcal{T}}(x,\mathbf{y})$ *be the highest scoring multilabel in* $\mathcal{Y}_{\mathcal{T},K}$. *Suppose that*

$$F_{\mathcal{T}}(x,\mathbf{y}_K^\star) \geq \frac{1}{n}\sum_{i=1}^{n} F_{\mathbf{w}_{T_i}}(x,\mathbf{y}_{T_i,K}) \stackrel{def}{=} \theta_x(K).$$

*It follows that* $F_{\mathcal{T}}(x,\mathbf{y}_K^\star) = \max_{\mathbf{y} \in \mathcal{Y}} F_{\mathcal{T}}(x,\mathbf{y})$.

We can use any $K$ satisfying the lemma as the length of $K$-best lists, and be assured that $\mathbf{y}_K^\star$ is a maximizer of $F_{\mathcal{T}}$.

We now examine the conditions under which the highest scoring multilabel is present in our candidate set $\mathcal{Y}_{\mathcal{T},K}$ with high probability. For any $x \in \mathcal{X}$ and any predictor $\mathbf{w}$, let $\hat{\mathbf{y}} \stackrel{def}{=} \mathbf{y}_{\mathbf{w}}(x) \stackrel{def}{=} \underset{\mathbf{y} \in \mathcal{Y}}{\operatorname{argmax}} F(\mathbf{w}, x, \mathbf{y})$ be the highest scoring multilabel in $\mathcal{Y}$ for predictor $\mathbf{w}$ on the complete graph $G$.

For any $\mathbf{y} \in \mathcal{Y}$, let $K_T(\mathbf{y})$ be the rank of $\mathbf{y}$ in tree $T$ and let $\rho_T(\mathbf{y}) \stackrel{def}{=} K_T(\mathbf{y})/|\mathcal{Y}|$ be the normalized rank of $\mathbf{y}$ in tree $T$. We then have $0 < \rho_T(\mathbf{y}) \leq 1$ and $\rho_T(\mathbf{y}') = \min_{\mathbf{y} \in \mathcal{Y}} \rho_T(\mathbf{y})$ whenever $\mathbf{y}'$ is a highest scoring multilabel in tree $T$. Since $\mathbf{w}$ and $x$ are arbitrary and fixed, let us drop them momentarily from the notation and let $F(\mathbf{y}) \stackrel{def}{=} F(\mathbf{w}, x, \mathbf{y})$, and $F_T(\mathbf{y}) \stackrel{def}{=} F_{\mathbf{w}_T}(x, \mathbf{y})$. Let $\mathcal{U}(\mathcal{Y})$ denote the uniform distribution of multilabels on $\mathcal{Y}$. Then, let $\mu_T \stackrel{def}{=} \mathbf{E}_{\mathbf{y} \sim \mathcal{U}(\mathcal{Y})} F_T(\mathbf{y})$ and $\mu \stackrel{def}{=} \mathbf{E}_{T \sim \mathcal{U}(G)} \mu_T$.

Let $\mathcal{T} \sim \mathcal{U}(G)^n$ be a sample of $n$ spanning trees of $G$. Since the scoring function $F_T$ of each tree $T$ of $G$ is bounded in absolute value, it follows that $F_T$ is a $\sigma_T$-sub-Gaussian random variable for some $\sigma_T > 0$. We now show that, with high probability, there exists a tree $T \in \mathcal{T}$ such that $\rho_T(\hat{\mathbf{y}})$ is decreasing exponentially rapidly with $(F(\hat{\mathbf{y}}) - \mu)/\sigma$, where $\sigma^2 \stackrel{def}{=} \mathbf{E}_{T \sim \mathcal{U}(G)} \sigma_T^2$.

**Lemma 8.** *Let the scoring function* $F_T$ *of each spanning tree of* $G$ *be a* $\sigma_T$-sub-Gaussian random variable under the uniform distribution of labels; i.e., for each $T$ on $G$, there exists $\sigma_T > 0$ such that for any $\lambda > 0$ we have

$$\underset{\mathbf{y} \sim \mathcal{U}(\mathcal{Y})}{\mathbf{E}} e^{\lambda(F_T(\mathbf{y}) - \mu_T)} \leq e^{\frac{\lambda^2}{2}\sigma_T^2}.$$

*Let* $\sigma^2 \stackrel{def}{=} \underset{T \sim \mathcal{U}(G)}{\mathbf{E}} \sigma_T^2$, *and let* $\alpha \stackrel{def}{=} \underset{T \sim \mathcal{U}(G)}{\operatorname{Pr}} \left( \mu_T \leq \mu \ \wedge \ F_T(\hat{\mathbf{y}}) \geq F(\hat{\mathbf{y}}) \ \wedge \ \sigma_T^2 \leq \sigma^2 \right)$. *Then,*

$$\underset{\mathcal{T} \sim \mathcal{U}(G)^n}{\operatorname{Pr}} \left( \exists T \in \mathcal{T} \colon \rho_T(\hat{\mathbf{y}}) \leq e^{-\frac{1}{2}\frac{(F(\hat{\mathbf{y}}) - \mu)^2}{\sigma^2}} \right) \geq 1 - (1 - \alpha)^n.$$

Thus, even for very small $\alpha$, when $n$ is large enough, there exists, with high probability, a tree $T \in \mathcal{T}$ such that $\hat{\mathbf{y}}$ has a small $\rho_T(\hat{\mathbf{y}})$ whenever $[F(\hat{\mathbf{y}}) - \mu]/\sigma$ is large for $G$. For example, when $|\mathcal{Y}| = 2^\ell$ (the multiple binary classification case), we have with probability of at least $1 - (1 - \alpha)^n$, that there exists $T \in \mathcal{T}$ such that $K_T(\hat{\mathbf{y}}) = 1$ whenever $F(\hat{\mathbf{y}}) - \mu \geq \sigma\sqrt{2\ell \ln 2}$.

### 4.2 Optimization

To optimize the $L_2$-norm RTA problem (Definition 6) we convert it to the marginalized dual form (see the supplementary material for the derivation), which gives us a polynomial-size problem (in the number of microlabels) and allows us to use kernels to tackle complex input spaces efficiently.

**Definition 9.** $L_2$-**norm RTA Marginalized Dual**

$$\underset{\boldsymbol{\mu} \in \mathcal{M}^m}{\boldsymbol{max}} \quad \frac{1}{|E_{\mathcal{T}}|} \sum_{e,k,\mathbf{u}_e} \mu(k,e,\mathbf{u}_e) - \frac{1}{2} \sum_{\substack{e,k,\mathbf{u}_e, \\ k',\mathbf{u}_e'}} \mu(k,e,\mathbf{u}_e) K_{\mathcal{T}}^e(x_k, \mathbf{u}_e; x_k', \mathbf{u}_e') \mu(k',e,\mathbf{u}_e'),$$

*where* $E_{\mathcal{T}}$ *is the union of the sets of edges appearing in* $\mathcal{T}$, *and* $\boldsymbol{\mu} \in \mathcal{M}^m$ *are the marginal dual variables* $\boldsymbol{\mu} \stackrel{def}{=} (\mu(k,e,\mathbf{u}_e))_{k,e,\mathbf{u}_e}$, *with the triplet* $(k,e,\mathbf{u}_e)$ *corresponding to labeling the edge*

| DATASET | MICROLABEL LOSS (%) | | | | | 0/1 LOSS (%) | | | | |
|---|---|---|---|---|---|---|---|---|---|---|
| | SVM | MTL | MMCRF | MAM | RTA | SVM | MTL | MMCRF | MAM | RTA |
| EMOTIONS | 22.4 | 20.2 | 20.1 | *19.5* | **18.8** | 77.8 | 74.5 | 71.3 | *69.6* | **66.3** |
| YEAST | *20.0* | 20.7 | 21.7 | 20.1 | **19.8** | *85.9* | 88.7 | 93.0 | 86.0 | **77.7** |
| SCENE | *9.8* | 11.6 | 18.4 | 17.0 | **8.8** | *47.2* | 55.2 | 72.2 | 94.6 | **30.2** |
| ENRON | 6.4 | 6.5 | 6.2 | **5.0** | *5.3* | 99.6 | 99.6 | 92.7 | *87.9* | **87.7** |
| CAL500 | **13.7** | *13.8* | **13.7** | **13.7** | *13.8* | 100.0 | 100.0 | 100.0 | 100.0 | 100.0 |
| FINGERPRINT | **10.3** | 17.3 | *10.5* | *10.5* | 10.7 | 99.0 | 100.0 | *99.6* | *99.6* | **96.7** |
| NCI60 | 15.3 | 16.0 | *14.6* | **14.3** | 14.9 | 56.9 | *53.0* | 63.1 | 60.0 | **52.9** |
| MEDICAL | *2.6* | *2.6* | **2.1** | **2.1** | **2.1** | 91.8 | 91.8 | 63.8 | *63.1* | **58.8** |
| CIRCLE10 | 4.7 | 6.3 | 2.6 | *2.5* | **0.6** | 28.9 | 33.2 | 20.3 | *17.7* | **4.0** |
| CIRCLE50 | 5.7 | 6.2 | **1.5** | *2.1* | 3.8 | 69.8 | 72.3 | **38.8** | *46.2* | 52.8 |

Table 1: Prediction performance of each algorithm in terms of microlabel loss and 0/1 loss. The best performing algorithm is highlighted with **boldface**, the second best is in *italic*.

$e = (v, v') \in E_{\mathcal{T}}$ *of the output graph by* $\mathbf{u}_e = (u_v, u_{v'}) \in \mathcal{Y}_v \times \mathcal{Y}_{v'}$ *for the training example* $x_k$*. Also,* $\mathcal{M}^m$ *is the marginal dual feasible set and*

$$K_{\mathcal{T}}^e(x_k, \mathbf{u}_e; x_{k'}, \mathbf{u}'_e) \stackrel{def}{=} \frac{N_{\mathcal{T}}(e)}{|E_{\mathcal{T}}|^2} K(x_k, x_{k'}) \left\langle \boldsymbol{\psi}_e(y_{kv}, y_{kv'}) - \boldsymbol{\psi}_e(u_v, u_{v'}), \boldsymbol{\psi}_e(y_{k'v}, y_{k'v'}) - \boldsymbol{\psi}_e(u'_v, u'_{v'}) \right\rangle$$

*is the joint kernel of input features and the differences of output features of true and competing multilabels* $(\mathbf{y}_k, \mathbf{u})$*, projected to the edge* $e$*. Finally,* $\mathcal{N}_{\mathcal{T}}(e)$ *denotes the number of times* $e$ *appears among the trees of the ensemble.*

The master algorithm described in the supplementary material iterates over each training example until convergence. The processing of each training example $x_k$ proceeds by finding the worst violating multilabel of the ensemble defined as

$$\bar{\mathbf{y}}_k \stackrel{def}{=} \underset{\mathbf{y} \neq \mathbf{y}_k}{\operatorname{argmax}} F_{\mathcal{T}}(x_k, \mathbf{y}), \tag{6}$$

using the $K$-best inference approach of the previous section, with the modification that the correct multilabel is excluded from the $K$-best lists. The worst violator $\bar{\mathbf{y}}_k$ is mapped to a vertex

$$\bar{\boldsymbol{\mu}}(x_k) = C \cdot \left( [\bar{\mathbf{y}}_e = \mathbf{u}_e] \right)_{e, \mathbf{u}_e} \in \mathcal{M}_k$$

corresponding to the steepest feasible ascent direction (*c.f*, [9]) in the marginal dual feasible set $\mathcal{M}_k$ of example $x_k$, thus giving us a subgradient of the objective of Definition 9. An exact line search is used to find the saddle point between the current solution and $\bar{\boldsymbol{\mu}}$.

## 5  Empirical Evaluation

We compare our method RTA to Support Vector Machine (SVM) [10, 11], Multitask Feature Learning (MTL) [12], Max-Margin Conditional Random Fields (MMCRF) [9] which uses the loopy belief propagation algorithm for approximate inference on the general graph, and Maximum Average Marginal Aggregation (MAM) [5] which is a multilabel ensemble model that trains a set of random tree based learners separately and performs the final approximate inference on a union graph of the edge potential functions of the trees. We use ten multilabel datasets from [5]. Following [5], MAM is constructed with 180 tree based learners, and for MMCRF a consensus graph is created by pooling edges from 40 trees. We train RTA with up to 40 spanning trees and with $K$ up to 32. The linear kernel is used for methods that require kernelized input. Margin slack parameters are selected from $\{100, 50, 10, 1, 0.5, 0.1, 0.01\}$. We use 5-fold cross-validation to compute the results.

**Prediction performance.** Table 1 shows the performance in terms of microlabel loss and 0/1 loss. The best methods are highlighted in '**boldface**' and the second best in '*italics*' (see supplementary material for full results). RTA quite often improves over MAM in 0/1 accuracy, sometimes with noticeable margin except for *Enron* and *Circle50*. The performances in microlabel accuracy are quite similar while RTA is slightly above the competition. This demonstrates the advantage of RTA that gains by optimizing on a collection of trees simultaneously rather than optimizing on individual trees as MAM. In addition, learning using approximate inference on a general graph seems less

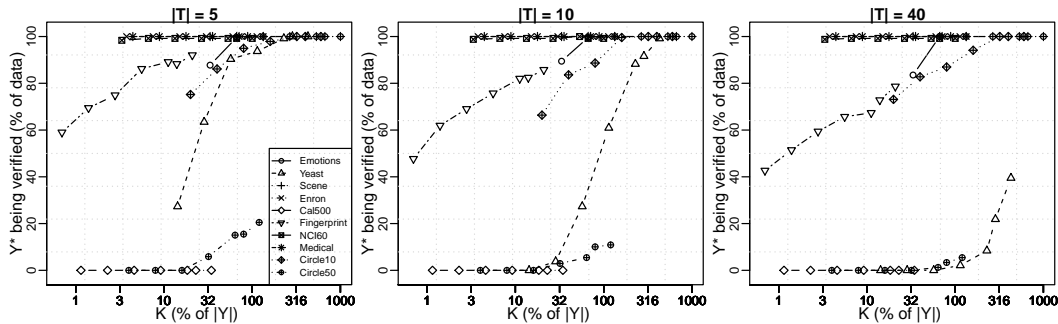

Figure 1: Percentage of examples with provably optimal $\mathbf{y}^*$ being in the $K$-best lists plotted as a function of $K$, scaled with respect to the number of microlabels in the dataset.

favorable as the tree-based methods, as MMCRF quite consistently trails to RTA and MAM in both microlabel and 0/1 error, except for *Circle50* where it outperforms other models. Finally, we notice that SVM, as a single label classifier, is very competitive against most multilabel methods for microlabel accuracy.

**Exactness of inference on the collection of trees.** We now study the empirical behavior of the inference (see Section 4) on the collection of trees, which, if taken as a single general graph, would call for solving an $\mathcal{NP}$-hard inference problem. We provide here empirical evidence that we can perform exact inference on most examples in most datasets in polynomial time.

We ran the $K$-best inference on eleven datasets where the RTA models were trained with different amounts of spanning trees $|\mathcal{T}| = \{5, 10, 40\}$ and values for $K = \{2, 4, 8, 16, 32, 40, 60\}$. For each parameter combination and for each example, we recorded whether the $K$-best inference was provably exact on the collection (*i.e.*, if Lemma 7 was satisfied). Figure 1 plots the percentage of examples where the inference was indeed provably exact. The values are shown as a function of $K$, expressed as the percentage of the number of microlabels in each dataset. Hence, $100\%$ means $K = \ell$, which denotes low polynomial ($\Theta(n\ell^2)$) time inference in the exponential size multilabel space.

We observe, from Figure 1, on some datasets (*e.g.*, *Medical*, *NCI60*), that the inference task is very easy since exact inference can be computed for most of the examples even with $K$ values that are below $50\%$ of the number of microlabels. By setting $K = \ell$ (*i.e.*, $100\%$) we can perform exact inference for about $90\%$ of the examples on nine datasets with five trees, and eight datasets with $40$ trees. On two of the datasets (*Cal500*, *Circle50*), inference is not (in general) exact with low values of $K$. Allowing $K$ to grow superlinearly on $\ell$ would possibly permit exact inference on these datasets. However, this is left for future studies.

Finally, we note that the difficulty of performing provably exact inference slightly increases when more spanning trees are used. We have observed that, in most cases, the optimal multilabel $\mathbf{y}^*$ is still on the $K$-best lists but the conditions of Lemma 7 are no longer satisfied, hence forbidding us to prove exactness of the inference. Thus, working to establish alternative proofs of exactness is a worthy future research direction.

## 6 Conclusion

The main theoretical result of the paper is the demonstration that if a large margin structured output predictor exists, then combining a small sample of random trees will, with high probability, generate a predictor with good generalization. The key attraction of this approach is the tractability of the inference problem for the ensemble of trees, both indicated by our theoretical analysis and supported by our empirical results. However, as a by-product, we have a significant added benefit: we do not need to know the output structure *a priori* as this is generated implicitly in the learned weights for the trees. This is used to significant advantage in our experiments that automatically leverage correlations between the multiple target outputs to give a substantive increase in accuracy. It also suggests that the approach has enormous potential for applications where the structure of the output is not known but is expected to play an important role.

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
