[Supplementary Material]

## Multilabel Structured Output Learning with Random Spanning Trees of Max-Margin Markov Networks (supplementary material) by: Mario Marchand, Hongyu Su, Emilie Morvant, Juho Rousou, and John Shawe-Taylor

## Lemma 1

*Let $\hat{\mathbf{w}}_T \stackrel{\text{def}}{=} \mathbf{w}_T/\|\mathbf{w}_T\|$, $\hat{\boldsymbol{\phi}}_T \stackrel{\text{def}}{=} \boldsymbol{\phi}_T/\|\boldsymbol{\phi}_T\|$. Let $\mathcal{U}(G)$ denote the uniform distribution on $S(G)$. Then, we have*

$$F(\mathbf{w}, x, \mathbf{y}) = \mathop{\mathbf{E}}_{T \sim \mathcal{U}(G)} a_T \langle \hat{\mathbf{w}}_T, \hat{\boldsymbol{\phi}}_T(x, \mathbf{y}) \rangle, \text{ where } a_T \stackrel{\text{def}}{=} \sqrt{\frac{\ell}{2}} \|\mathbf{w}_T\|.$$

*Moreover, for any $\mathbf{w}$ such that $\|\mathbf{w}\| = 1$, we have*

$$\mathop{\mathbf{E}}_{T \sim \mathcal{U}(G)} a_T^2 = 1 \quad ; \quad \mathop{\mathbf{E}}_{T \sim \mathcal{U}(G)} a_T \le 1.$$

*Proof.*

$$
\begin{aligned}
F(\mathbf{w}, x, \mathbf{y}) &= \langle \mathbf{w}, \boldsymbol{\phi}(x, \mathbf{y}) \rangle \\
&= \sum_{(i,j) \in G} \langle \mathbf{w}_{i,j}, \boldsymbol{\phi}_{i,j}(x, y_i, y_j) \rangle = \frac{1}{\ell^{\ell-2}} \frac{\ell}{2} \sum_{T \in S(G)} \sum_{(i,j) \in T} \langle \mathbf{w}_{i,j}, \boldsymbol{\phi}_{i,j}(x, y_i, y_j) \rangle \\
&= \frac{\ell}{2} \mathop{\mathbf{E}}_{T \sim \mathcal{U}(G)} \langle \mathbf{w}_T, \boldsymbol{\phi}_T(x, \mathbf{y}) \rangle = \frac{\ell}{2} \mathop{\mathbf{E}}_{T \sim \mathcal{U}(G)} \|\mathbf{w}_T\| \|\boldsymbol{\phi}_T(x, \mathbf{y})\| \langle \hat{\mathbf{w}}_T, \hat{\boldsymbol{\phi}}_T(x, \mathbf{y}) \rangle \\
&= \mathop{\mathbf{E}}_{T \sim \mathcal{U}(G)} \sqrt{\frac{\ell}{2}} \|\mathbf{w}_T\| \langle \hat{\mathbf{w}}_T, \hat{\boldsymbol{\phi}}_T(x, \mathbf{y}) \rangle = \mathop{\mathbf{E}}_{T \sim \mathcal{U}(G)} a_T \langle \hat{\mathbf{w}}_T, \hat{\boldsymbol{\phi}}_T(x, \mathbf{y}) \rangle,
\end{aligned}
$$

where

$$a_T \stackrel{\text{def}}{=} \sqrt{\frac{\ell}{2}} \|\mathbf{w}_T\|.$$

Now, for any $\mathbf{w}$ such that $\|\mathbf{w}\| = 1$, we have

$$
\begin{aligned}
\mathop{\mathbf{E}}_{T \sim \mathcal{U}(G)} a_T^2 &= \frac{\ell}{2} \mathop{\mathbf{E}}_{T \sim \mathcal{U}(G)} \|\mathbf{w}_T\|^2 = \frac{\ell}{2} \frac{1}{\ell^{\ell-2}} \sum_{T \in S(G)} \|\mathbf{w}_T\|^2 = \frac{\ell}{2} \frac{1}{\ell^{\ell-2}} \sum_{T \in S(G)} \sum_{(i,j) \in T} \|\mathbf{w}_{i,j}\|^2 \\
&= \sum_{(i,j) \in G} \|\mathbf{w}_{i,j}\|^2 = \|\mathbf{w}\|^2 = 1.
\end{aligned}
$$

Since the variance of $a_T$ must be positive, we have, for any $\mathbf{w}$ of unit $L_2$ norm, that

$$\mathop{\mathbf{E}}_{T \sim \mathcal{U}(G)} a_T \le 1.$$

$\square$

## Lemma 2

*Consider any unit $L_2$ norm predictor $\mathbf{w}$ on the complete graph $G$ that achieves a margin of $\Gamma(\mathbf{w}, x, \mathbf{y})$ for each $(x, \mathbf{y}) \in \mathcal{X} \times \mathcal{Y}$, then we have*

$$\Gamma_{\mathcal{T}}(\mathbf{w}, x, \mathbf{y}) \ge \Gamma(\mathbf{w}, x, \mathbf{y}) - 2\epsilon \quad \forall(x, \mathbf{y}) \in \mathcal{X} \times \mathcal{Y},$$

*whenever for all $(x, \mathbf{y}) \in \mathcal{X} \times \mathcal{Y}$, we have*

$$|F_{\mathcal{T}}(\mathbf{w}, x, \mathbf{y}) - F(\mathbf{w}, x, \mathbf{y})| \le \epsilon.$$

*Proof.* From the condition of the lemma, we have simultaneously for all $(x, \mathbf{y}) \in \mathcal{X} \times \mathcal{Y}$ and $(x, \mathbf{y}') \in \mathcal{X} \times \mathcal{Y}$, that

$$F_{\mathcal{T}}(\mathbf{w}, x, \mathbf{y}) \geq F(\mathbf{w}, x, \mathbf{y}) - \epsilon \quad \text{AND} \quad F_{\mathcal{T}}(\mathbf{w}, x, \mathbf{y}') \leq F(\mathbf{w}, x, \mathbf{y}') + \epsilon \,.$$

Therefore,

$$F_{\mathcal{T}}(\mathbf{w}, x, \mathbf{y}) - F_{\mathcal{T}}(\mathbf{w}, x, \mathbf{y}') \geq F(\mathbf{w}, x, \mathbf{y}) - F(\mathbf{w}, x, \mathbf{y}') - 2\epsilon \,.$$

Hence, for all $(x, \mathbf{y}) \in \mathcal{X} \times \mathcal{Y}$, we have

$$\Gamma_{\mathcal{T}}(\mathbf{w}, x, \mathbf{y}) \geq \Gamma(\mathbf{w}, x, \mathbf{y}) - 2\epsilon \,.$$

$\square$

## Lemma 3

*Consider any $\epsilon > 0$ and any unit $L_2$ norm predictor $\mathbf{w}$ for the complete graph $G$ acting on a normalized joint feature space. For any $\delta \in (0, 1)$, let*

$$n \geq \frac{\ell^2}{\epsilon^2} \left( \frac{1}{16} + \frac{1}{2} \ln \frac{8\sqrt{n}}{\delta} \right)^2 \,. \tag{2}$$

*Then with probability of at least $1 - \delta/2$ over all samples $\mathcal{T}$ generated according to $\mathcal{U}(G)^n$, we have, simultaneously for all $(x, \mathbf{y}) \in \mathcal{X} \times \mathcal{Y}$, that*

$$|F_{\mathcal{T}}(\mathbf{w}, x, \mathbf{y}) - F(\mathbf{w}, x, \mathbf{y})| \leq \epsilon \,.$$

*Proof.* Consider an isotropic Gaussian distribution of joint feature vectors of variance $\sigma^2$, centred on $\boldsymbol{\phi}(x, \mathbf{y})$, with a density given by

$$Q_{\boldsymbol{\phi}}(\boldsymbol{\zeta}) \stackrel{\text{def}}{=} \left( \frac{1}{\sqrt{2\pi}\sigma} \right)^N \exp - \frac{\|\boldsymbol{\zeta} - \boldsymbol{\phi}\|^2}{2\sigma^2} \,,$$

where $N$ is the dimension of the feature vectors. When the feature space is infinite-dimensional, we can consider $Q$ to be a Gaussian process. The end results will not depend on $N$.

Given the fixed $\mathbf{w}$ stated in the theorem, let us define the risk $R(Q_{\boldsymbol{\phi}}, \mathbf{w}_T)$ of $Q_{\boldsymbol{\phi}}$ on the tree $T$ by $\underset{\boldsymbol{\zeta} \sim Q_{\boldsymbol{\phi}}}{\mathbf{E}} \langle \mathbf{w}_T, \boldsymbol{\zeta} \rangle$. By the linearity of $\langle \cdot, \cdot \rangle$, we have

$$R(Q_{\boldsymbol{\phi}}, \mathbf{w}_T) \stackrel{\text{def}}{=} \underset{\boldsymbol{\zeta} \sim Q_{\boldsymbol{\phi}}}{\mathbf{E}} \langle \mathbf{w}_T, \boldsymbol{\zeta} \rangle = \langle \mathbf{w}_T, \underset{\boldsymbol{\zeta} \sim Q_{\boldsymbol{\phi}}}{\mathbf{E}} \boldsymbol{\zeta} \rangle = \langle \mathbf{w}_T, \boldsymbol{\phi} \rangle \,,$$

which is independent of $\sigma$.

Gibbs' risk $R(Q_{\boldsymbol{\phi}})$ and its empirical estimate $R_{\mathcal{T}}(Q_{\boldsymbol{\phi}})$ are defined as

$$R(Q_{\boldsymbol{\phi}}) \stackrel{\text{def}}{=} \underset{T \sim \mathcal{U}(G)}{\mathbf{E}} R(Q_{\boldsymbol{\phi}}, \mathbf{w}_T) = \underset{T \sim \mathcal{U}(G)}{\mathbf{E}} \langle \mathbf{w}_T, \boldsymbol{\phi} \rangle$$

$$R_{\mathcal{T}}(Q_{\boldsymbol{\phi}}) \stackrel{\text{def}}{=} \frac{1}{n} \sum_{i=1}^{n} R(Q_{\boldsymbol{\phi}}, \mathbf{w}_{T_i}) = \frac{1}{n} \sum_{i=1}^{n} \langle \mathbf{w}_{T_i}, \boldsymbol{\phi} \rangle \,.$$

Consequently, from the definitions of $F$ and $F_{\mathcal{T}}$, we have

$$F(\mathbf{w}, x, \mathbf{y}) = \frac{\ell}{2} R(Q_{\boldsymbol{\phi}(x, \mathbf{y})})$$

$$F_{\mathcal{T}}(\mathbf{w}, x, \mathbf{y}) = \frac{\ell}{2} R_{\mathcal{T}}(Q_{\boldsymbol{\phi}(x, \mathbf{y})}) \,.$$

Recall that $\boldsymbol{\phi}$ is a normalized feature map that applies to all $(x, \mathbf{y}) \in \mathcal{X} \times \mathcal{Y}$. Therefore, if we have with probability $\geq 1 - \delta/2$ that, simultaneously for all $\boldsymbol{\phi}$ of unit $L_2$ norm,

$$\frac{\ell}{2} |R_{\mathcal{T}}(Q_{\boldsymbol{\phi}}) - R(Q_{\boldsymbol{\phi}})| \leq \epsilon \,, \tag{7}$$

then, with the same probability, we will have simultaneously $\forall (x, \mathbf{y}) \in \mathcal{X} \times \mathcal{Y}$, that

$$|F_{\mathcal{T}}(\mathbf{w}, x, \mathbf{y}) - F(\mathbf{w}, x, \mathbf{y})| \le \epsilon \,,$$

and, consequently, the lemma will be proved.

To prove that we satisfy Equation (7) with probability $\ge 1 - \delta/2$ simultaneously for all $\boldsymbol{\phi}$ of unit $L_2$ norm, let us adapt some elements of PAC-Bayes theory to our case. Note that we cannot use the usual PAC-Bayes bounds, such as those proposed by [13] because, in our case, the loss $\langle \mathbf{w}_T, \boldsymbol{\zeta} \rangle$ of each individual "predictor" $\boldsymbol{\zeta}$ is unbounded.

The distribution $Q_{\boldsymbol{\phi}}$ defined above constitutes the posterior distribution. For the prior $P$, let us use an isotropic Gaussian with variance $\sigma^2$ centered at the origin. Hence $P = Q_{\mathbf{0}}$. In that case we have

$$\mathrm{KL}(Q_{\boldsymbol{\phi}} \| P) = \frac{\|\boldsymbol{\phi}\|^2}{2\sigma^2} = \frac{1}{2\sigma^2} \,.$$

Given a tree sample $\mathcal{T}$ of $n$ spanning trees, let

$$\Delta \mathbf{w} \stackrel{\text{def}}{=} \frac{1}{n} \sum_{k=1}^{n} \mathbf{w}_{T_k} - \mathop{\mathbf{E}}_{T \sim \mathcal{U}(G)} \mathbf{w}_T \,,$$

and consider the Gaussian quadrature

$$\begin{aligned}
\mathcal{I} \stackrel{\text{def}}{=} \; & \mathop{\mathbf{E}}_{\boldsymbol{\zeta} \sim P} e^{\sqrt{n} |\langle \Delta \mathbf{w}, \boldsymbol{\zeta} \rangle|} \\
= \; & e^{\frac{1}{2} n \sigma^2 \|\Delta \mathbf{w}\|^2} \left( 1 + \mathrm{Erf} \left[ \sqrt{\frac{n}{2}} \|\Delta \mathbf{w}\| \sigma \right] \right) \\
\le \; & 2 e^{\frac{1}{2} n \sigma^2 \|\Delta \mathbf{w}\|^2} \,.
\end{aligned}$$

We can then use this result for $\mathcal{I}$ to upper bound the Laplace transform $\mathcal{L}$ in the following way.

$$\begin{aligned}
\mathcal{L} \stackrel{\text{def}}{=} \; & \mathop{\mathbf{E}}_{\mathcal{T} \sim \mathcal{U}(G)^n} \mathop{\mathbf{E}}_{\boldsymbol{\zeta} \sim P} e^{\sqrt{n} |\langle \Delta \mathbf{w}, \boldsymbol{\zeta} \rangle|} \\
\le \; & 2 \mathop{\mathbf{E}}_{\mathcal{T} \sim \mathcal{U}(G)^n} e^{\frac{1}{2} n \sigma^2 \|\Delta \mathbf{w}\|^2} \\
= \; & 2 \mathop{\mathbf{E}}_{\mathcal{T} \sim \mathcal{U}(G)^n} e^{\frac{1}{2} n \sigma^2 \sum_{(i,j) \in G} \|(\Delta \mathbf{w})_{i,j}\|^2} \,.
\end{aligned}$$

Since

$$\mathop{\mathbf{E}}_{T \sim \mathcal{U}(G)} \mathbf{w}_T = \frac{2}{\ell} \mathbf{w} \,,$$

we can write

$$\|(\Delta \mathbf{w})_{i,j}\|^2 = \left\| \frac{1}{n} \sum_{k=1}^{n} (\mathbf{w}_{T_k})_{i,j} - \frac{2}{\ell} \mathbf{w}_{i,j} \right\|^2 \,.$$

Note that for each $(i, j) \in G$, any sample $\mathcal{T}$, and each $T_k \in \mathcal{T}$, we can write

$$(\mathbf{w}_{T_k})_{i,j} = \mathbf{w}_{i,j} Z_{i,j}^k \,.$$

where $Z_{i,j}^k = 1$ if $(i, j) \in T_k$ and $Z_{i,j}^k = 0$ if $(i, j) \notin T_k$. Hence, we have

$$\|(\Delta \mathbf{w})_{i,j}\|^2 = \|\mathbf{w}_{i,j}\|^2 \left( \frac{1}{n} \sum_{k=1}^{n} Z_{i,j}^k - \frac{2}{\ell} \right)^2 \,.$$

Hence, for $\sigma^2 \le 4$ and $p \stackrel{\text{def}}{=} 2/\ell$, we have

$$\begin{aligned}
\mathcal{L} \le \; & 2 \mathop{\mathbf{E}}_{\mathcal{T} \sim \mathcal{U}(G)^n} e^{\frac{1}{2} n \sigma^2 \sum_{(i,j) \in G} \|\mathbf{w}_{i,j}\|^2 \left( \frac{1}{n} \sum_{k=1}^{n} Z_{i,j}^k - \frac{2}{\ell} \right)^2} \\
\le \; & 2 \mathop{\mathbf{E}}_{\mathcal{T} \sim \mathcal{U}(G)^n} e^{2n \sum_{(i,j) \in G} \|\mathbf{w}_{i,j}\|^2 \left( \frac{1}{n} \sum_{k=1}^{n} Z_{i,j}^k - p \right)^2} \\
\le \; & 2 \sum_{(i,j) \in G} \|\mathbf{w}_{i,j}\|^2 \mathop{\mathbf{E}}_{\mathcal{T} \sim \mathcal{U}(G)^n} e^{2n \left( \frac{1}{n} \sum_{k=1}^{n} Z_{i,j}^k - p \right)^2} \,,
\end{aligned}$$

where the last inequality is obtained by using $\sum_{(i,j)\in G} \|\mathbf{w}_{i,j}\|^2 = 1$ and by using Jensen's inequality on the convexity of the exponential.

Now, for any $(q,p) \in [0,1]^2$, let

$$\mathrm{kl}(q\|p) \overset{\text{def}}{=} q\ln\frac{q}{p} + (1-q)\ln\frac{1-q}{1-p} .$$

Then, by using $2(q-p)^2 \leq \mathrm{kl}(q\|p)$ (Pinsker's inequality), we have for $n \geq 8$,

$$\mathcal{L} \ \leq \ 2\sum_{(i,j)\in G} \|\mathbf{w}_{i,j}\|^2 \underset{\mathcal{T}\sim\mathcal{U}(G)^n}{\mathbf{E}} e^{n\mathrm{kl}\left(\frac{1}{n}\sum_{k=1}^n Z_{i,j}^k \| p\right)} \ \leq \ 4\sqrt{n} ,$$

where the last inequality follows from Maurer's lemma [14] applied, for any fixed $(i,j) \in G$, to the collection of $n$ independent Bernoulli variables $Z_{i,j}^k$ of probability $p$.

The rest of the proof follows directly from standard PAC-Bayes theory [15, 13], which, for completeness, we briefly outline here.

Since

$$\underset{\boldsymbol{\zeta}\sim P}{\mathbf{E}}\ e^{\sqrt{n}|\langle\Delta\mathbf{w},\boldsymbol{\zeta}\rangle|}$$

is a non negative random variable, Markov's inequality implies that with probability $> 1 - \delta/2$ over the random draws of $\mathcal{T}$, we have

$$\ln\underset{\boldsymbol{\zeta}\sim P}{\mathbf{E}}\ e^{\sqrt{n}|\langle\Delta\mathbf{w},\boldsymbol{\zeta}\rangle|} \ \leq \ln\frac{8\sqrt{n}}{\delta} .$$

By the change of measure inequality, we have with probability $> 1 - \delta/2$ over the random draws of $\mathcal{T}$, simultaneously for all $\boldsymbol{\phi}$,

$$\sqrt{n}\underset{\boldsymbol{\zeta}\sim Q_{\boldsymbol{\phi}}}{\mathbf{E}}\ |\langle\Delta\mathbf{w},\boldsymbol{\zeta}\rangle| \ \leq \ \mathrm{KL}\left(Q_{\boldsymbol{\phi}}\|P\right) + \ln\frac{8\sqrt{n}}{\delta} .$$

Hence, by using Jensen's inequality on the convex absolute value function, we have with probability $> 1 - \delta/2$ over the random draws of $\mathcal{T}$, simultaneously for all $\boldsymbol{\phi}$,

$$|\langle\Delta\mathbf{w},\boldsymbol{\phi}\rangle| \ \leq \ \frac{1}{\sqrt{n}}\left[\mathrm{KL}\left(Q_{\boldsymbol{\phi}}\|P\right) + \ln\frac{8\sqrt{n}}{\delta}\right] .$$

Note that we have $\mathrm{KL}(Q_{\boldsymbol{\phi}}\|P) = 1/8$ for $\sigma^2 = 4$ (which is the value we shall use). Also note that the left hand side of this equation equals to $|R_{\mathcal{T}}(Q_{\boldsymbol{\phi}}) - R(Q_{\boldsymbol{\phi}})|$. In that case, we satisfy Equation (7) with probability $1 - \delta/2$ simultaneously for all $\boldsymbol{\phi}$ of unit $L_2$ norm whenever we satisfy

$$\frac{\ell}{2\sqrt{n}}\left[\frac{1}{8} + \ln\frac{8\sqrt{n}}{\delta}\right] \ \leq \ \epsilon ,$$

which is the condition on $n$ given by the theorem. $\qquad\square$

## Theorem 4

*Consider any unit $L_2$ norm predictor $\mathbf{w}$ for the complete graph $G$, acting on a normalized joint feature space, achieving a margin of $\Gamma(\mathbf{w}, x, \mathbf{y})$ for each $(x, \mathbf{y}) \in \mathcal{X} \times \mathcal{Y}$. Then for any $\epsilon > 0$, and any $n$ satisfying Lemma 3, for any $\delta \in (0,1]$, with probability of at least $1 - \delta$ over all samples $\mathcal{T}$ generated according to $\mathcal{U}(G)^n$, there exists a unit $L_2$ norm conical combination $(\mathcal{W}, \mathbf{q})$ on $\mathcal{T}$ such that, simultaneously $\forall(x, \mathbf{y}) \in \mathcal{X} \times \mathcal{Y}$, we have*

$$\Gamma_{\mathcal{T}}(\mathcal{W}, \mathbf{q}, x, \mathbf{y}) \geq \frac{1}{\sqrt{1+\epsilon}}\left[\Gamma(\mathbf{w}, x, \mathbf{y}) - 2\epsilon\right] .$$

*Proof.* For any $\mathcal{T}$, consider a conical combination $(\mathcal{W}, \mathbf{q})$ where each $\hat{\mathbf{w}}_{T_i} \in \mathcal{W}$ is obtained by projecting $\mathbf{w}$ on $T_i$ and normalizing to unit $L_2$ norm and where

$$q_i = \frac{a_{T_i}}{\sqrt{\sum_{i=1}^{n} a_{T_i}^2}}.$$

Then, from equations (3) and (4), and from the definition of $\Gamma_{\mathcal{T}}(\mathbf{w}, x, \mathbf{y})$, we find that for all $(x, \mathbf{y}) \in \mathcal{X} \times \mathcal{Y}$, we have

$$\Gamma_{\mathcal{T}}(\mathcal{W}, \mathbf{q}, x, \mathbf{y}) = \sqrt{\frac{n}{\sum_{i=1}^{n} a_{T_i}^2}} \Gamma_{\mathcal{T}}(\mathbf{w}, x, \mathbf{y}).$$

Now, by using Hoeffding's inequality, it is straightforward to show that for any $\delta \in (0, 1]$, we have

$$\Pr_{\mathcal{T} \sim \mathcal{U}(G)^n} \left( \frac{1}{n} \sum_{i=1}^{n} a_{T_i}^2 \leq 1 + \epsilon \right) \geq 1 - \delta/2.$$

whenever $n \geq \frac{\ell^2}{8\epsilon} \ln\left(\frac{2}{\delta}\right)$. Since $n$ satisfies the condition of Lemma 3, we see that it also satisfies this condition whenever $\epsilon \leq 1/2$. Hence, with probability of at least $1 - \delta/2$, we have

$$\sum_{i=1}^{n} a_{T_i}^2 \leq n(1 + \epsilon).$$

Moreover Lemma 2 and Lemma 3 imply that, with probability of at least $1 - \delta/2$, we have simultaneously for all $(x, \mathbf{y}) \in \mathcal{X} \times \mathcal{Y}$,

$$\Gamma_{\mathcal{T}}(\mathbf{w}, x, \mathbf{y}) \geq \Gamma(\mathbf{w}, x, \mathbf{y}) - 2\epsilon.$$

Hence, from the union bound, with probability of at least $1 - \delta$, simultaneously $\forall (x, \mathbf{y}) \in \mathcal{X} \times \mathcal{Y}$, we have

$$\Gamma_{\mathcal{T}}(\mathcal{W}, \mathbf{q}, x, \mathbf{y}) \geq \frac{1}{\sqrt{1+\epsilon}} \left[ \Gamma(\mathbf{w}, x, \mathbf{y}) - 2\epsilon \right].$$

$\square$

## Derivation of the Primal $L_2$-norm Random Tree Approximation

If we introduce the usual slack variables $\xi_i \stackrel{\text{def}}{=} \gamma \cdot \mathcal{L}^{\gamma}(\Gamma_{\mathcal{T}}(\mathcal{W}, \mathbf{q}, x_i, \mathbf{y}_i))$, Theorem 5 suggests that we should minimize $\frac{1}{\gamma} \sum_{k=1}^{m} \xi_k$ for some fixed margin value $\gamma > 0$. Rather than performing this task for several values of $\gamma$, we can, equivalently, solve the following optimization problem for several values of $C > 0$.

$$\min_{\boldsymbol{\xi}, \gamma, \mathbf{q}, \mathcal{W}} \quad \frac{1}{2\gamma^2} + \frac{C}{\gamma} \sum_{k=1}^{m} \xi_k \tag{8}$$

$$\text{s.t.}: \quad \Gamma_{\mathcal{T}}(\mathcal{W}, \mathbf{q}, x_k, \mathbf{y}_k) \geq \gamma - \xi_k, \ \xi_k \geq 0, \ \forall k \in \{1, \dots, m\},$$

$$\sum_{i=1}^{n} q_i^2 = 1, \ q_i \geq 0, \ \|\mathbf{w}_{T_i}\|^2 = 1, \ \forall i \in \{1, \dots, n\}.$$

If we now use instead $\zeta_k \stackrel{\text{def}}{=} \xi_k/\gamma$, and $\mathbf{v}_{T_i} \stackrel{\text{def}}{=} q_i \mathbf{w}_{T_i}/\gamma$, we then have $\sum_{i=1}^{n} \|\mathbf{v}_{T_i}\|^2 = 1/\gamma^2$ (under the constraints of problem (8)). If $\mathcal{V} \stackrel{\text{def}}{=} \{\mathbf{v}_{T_1}, \dots, \mathbf{v}_{T_n}\}$, optimization problem (8) is then equivalent to

$$\min_{\boldsymbol{\zeta}, \mathcal{V}} \quad \frac{1}{2} \sum_{i=1}^{n} \|\mathbf{v}_{T_i}\|^2 + C \sum_{k=1}^{m} \zeta_k \tag{9}$$

$$\text{s.t.}: \quad \Gamma_{\mathcal{T}}(\mathcal{V}, \mathbf{1}, x_k, \mathbf{y}_k) \geq 1 - \zeta_k, \ \zeta_k \geq 0, \ \forall k \in \{1, \dots, m\}.$$

Note that, following our definitions, we now have

$$\Gamma_{\mathcal{T}}(\mathcal{V}, \mathbf{1}, x, \mathbf{y}) = \frac{1}{\sqrt{n}} \sum_{i=1}^{n} \langle \mathbf{v}_{T_i}, \hat{\boldsymbol{\phi}}_{T_i}(x, \mathbf{y}) \rangle - \max_{\mathbf{y}' \neq \mathbf{y}} \frac{1}{\sqrt{n}} \sum_{i=1}^{n} \langle \mathbf{v}_{T_i}, \hat{\boldsymbol{\phi}}_{T_i}(x, \mathbf{y}') \rangle.$$

We then obtain the optimization problem of Property 6 with the change of variables $\mathbf{w}_{T_i} \leftarrow \mathbf{v}_{T_i}/\sqrt{n}$, $\xi_k \leftarrow \zeta_k$, and $C \leftarrow C/\sqrt{n}$.

## Lemma 7

*Let $\mathbf{y}_K^{\star} = \underset{\mathbf{y} \in \mathcal{Y}_{\mathcal{T},K}}{\operatorname{argmax}} F_{\mathcal{T}}(x, \mathbf{y})$ be the highest scoring multilabel in $\mathcal{Y}_{\mathcal{T},K}$. Suppose that*

$$F_{\mathcal{T}}(x, \mathbf{y}_K^{\star}) \geq \frac{1}{n} \sum_{i=1}^{n} F_{\mathbf{w}_{T_i}}(x, \mathbf{y}_{T_i,K}) \overset{def}{=} \theta_x(K)$$

*It follows that $F_{\mathcal{T}}(x, \mathbf{y}_K^{\star}) = \max_{\mathbf{y} \in \mathcal{Y}} F_{\mathcal{T}}(x, \mathbf{y})$.*

*Proof.* Consider a multilabel $\mathbf{y}^{\dagger} \notin \mathcal{Y}_{\mathcal{T},K}$. It follows that for all $T_i$ we have

$$F_{\mathbf{w}_{T_i}}(x, \mathbf{y}^{\dagger}) \leq F_{\mathbf{w}_{T_i}}(x, \mathbf{y}_{T_i,K}).$$

Hence,

$$F_{\mathcal{T}}(x, \mathbf{y}^{\dagger}) \;=\; \frac{1}{n} \sum_{i=1}^{n} F_{\mathbf{w}_{T_i}}(x, \mathbf{y}^{\dagger}) \;\leq\; \frac{1}{n} \sum_{i=1}^{n} F_{\mathbf{w}_{T_i}}(x, \mathbf{y}_{T_i,K}) \;\leq\; F_{\mathcal{T}}(x, \mathbf{y}_K^{\star}),$$

as required. $\qquad\square$

## Lemma 8

*Let the scoring function $F_T$ of each spanning tree of $G$ be a $\sigma_T$-sub-Gaussian random variable under the uniform distribution of labels; i.e., for each $T$ on $G$, there exists $\sigma_T > 0$ such that for any $\lambda > 0$ we have*

$$\underset{\mathbf{y} \sim \mathcal{U}(\mathcal{Y})}{\mathbf{E}} e^{\lambda(F_T(\mathbf{y}) - \mu_T)} \;\leq\; e^{\frac{\lambda^2}{2}\sigma_T^2}.$$

*Let $\sigma^2 \overset{def}{=} \underset{T \sim \mathcal{U}(G)}{\mathbf{E}} \sigma_T^2$, and let*

$$\alpha \overset{def}{=} \underset{T \sim \mathcal{U}(G)}{\operatorname{Pr}} \left( \mu_T \leq \mu \;\wedge\; F_T(\hat{\mathbf{y}}) \geq F(\hat{\mathbf{y}}) \;\wedge\; \sigma_T^2 \leq \sigma^2 \right).$$

*Then*

$$\underset{\mathcal{T} \sim \mathcal{U}(G)^n}{\operatorname{Pr}} \left( \exists T \in \mathcal{T}\colon \rho_T(\hat{\mathbf{y}}) \leq e^{-\frac{1}{2}\frac{(F(\hat{\mathbf{y}}) - \mu)^2}{\sigma^2}} \right) \geq 1 - (1 - \alpha)^n.$$

*Proof.* From the definition of $\rho(\hat{\mathbf{y}})$ and for any $\lambda > 0$, we have

$$
\begin{aligned}
\rho_T(\mathbf{y}^*) \;&=\; \underset{\mathbf{y} \sim \mathcal{U}(\mathcal{Y})}{\operatorname{Pr}} \left( F_T(\mathbf{y}) \geq F_T(\hat{\mathbf{y}}) \right) \\
&=\; \underset{\mathbf{y} \sim \mathcal{U}(\mathcal{Y})}{\operatorname{Pr}} \left( F_T(\mathbf{y}) - \mu_T \geq F_T(\hat{\mathbf{y}}) - \mu_T \right) \\
&=\; \underset{\mathbf{y} \sim \mathcal{U}(\mathcal{Y})}{\operatorname{Pr}} \left( e^{\lambda(F_T(\mathbf{y}) - \mu_T)} \geq e^{\lambda(F_T(\hat{\mathbf{y}}) - \mu_T)} \right) \\
&\leq\; e^{-\lambda(F_T(\hat{\mathbf{y}}) - \mu_T)} \underset{\mathbf{y} \sim \mathcal{U}(\mathcal{Y})}{\mathbf{E}} e^{\lambda(F_T(\mathbf{y}) - \mu_T)} & (10) \\
&\leq\; e^{-\lambda(F_T(\hat{\mathbf{y}}) - \mu_T)} e^{\frac{\lambda^2}{2}\sigma_T^2}, & (11)
\end{aligned}
$$

where we have used Markov's inequality for line (10) and the fact that $F_T$ is a $\sigma_T$-sub-Gaussian variable for line (11). Hence, from this equation and from the definition of $\alpha$, we have that

$$\underset{T \sim \mathcal{U}(G)}{\operatorname{Pr}} \left( \rho_T(\hat{\mathbf{y}}) \leq e^{-\lambda(F_T(\hat{\mathbf{y}}) - \mu_T)} e^{\frac{\lambda^2}{2}\sigma_T^2} \leq e^{-\lambda(F(\hat{\mathbf{y}}) - \mu)} e^{\frac{\lambda^2}{2}\sigma^2} \right) \;\geq\; \alpha.$$

Hence,

$$\underset{\mathcal{T} \sim \mathcal{U}(G)^n}{\operatorname{Pr}} \left( \forall T \in \mathcal{T}\colon \rho_T(\hat{\mathbf{y}}) > e^{-\lambda(F(\hat{\mathbf{y}}) - \mu)} e^{\frac{\lambda^2}{2}\sigma^2} \right) \;\leq\; (1 - \alpha)^n,$$

which is equivalent to the statement of the lemma when we choose $\lambda = [F(\hat{\mathbf{y}}) - \mu]/\sigma^2$. $\qquad\square$

## The $K$-best Inference Algorithm

Algorithm 1 depicts the $K$-best inference algorithm for the ensemble of rooted spanning trees. The algorithm takes as input the collection of spanning trees $T_i \in \mathcal{T}$, the edge labeling scores

$$F_{E_\mathcal{T}} = \{F_{T_i,v,v'}(y_v, y_{v'})\}_{(v,v') \in E_i, y_v \in \mathcal{Y}_v, y_{v'} \in \mathcal{Y}_{v'}, T_i \in \mathcal{T}}$$

for fixed $x_k$ and $\mathbf{w}$, the length of $K$-best list, and optionally (for training) also the true multilabel $\mathbf{y}_k$ for $x_k$.

As a rooted tree implicitly orients the edges, for convenience we denote the edges as directed $v \rightarrow pa(v)$, where $pa(v)$ denotes the parent (i.e. the adjacent node on the path towards the root) of $v$. By $ch(v)$ we denote the set of children of $v$. Moreover, we denote the subtree of $T_i$ rooted at a node $v$ as $T_v$ and by $T_{v' \rightarrow v}$ the subtree consisting of $T_{v'}$ plus the edge $v' \rightarrow v$ and the node $v$.

The algorithm performs a dynamic programming over each tree in turn, extracting the $K$-best list of multilabels and their scores, and aggregates the results of the trees, retrieving the highest scoring multilabel of the ensemble, the worst violating multilabel and the threshold score of the $K$-best lists.

The dynamic programming is based on traversing the tree in post-order, so that children of the node are always processed before the parent. The algorithm maintains sorted $K$ best lists of candidate labelings of the subtrees $T_v$ and $T_{v' \rightarrow v}$, using the following data structures:

- Score matrix $P_v$, where element $P_v(y, r)$ records the score of the $r$'th best multilabel of the subtree $T_v$ when node $v$ is labeled as $y$.

- Pointer matrix $C_v$, where element $C_v(y, r)$ keeps track of the ranks of the child nodes $v' \in ch(v)$ in the message matrix $M_{v' \rightarrow v}$ that contributes to the score $P_v(y, r)$.

- Message matrix $M_{v \rightarrow pa(v)}$, where element $M_{v \rightarrow pa(v)}(y', r)$ records the score of $r$'th best multilabel of the subtree $T_{v \rightarrow pa(v)}$ when the label of $pa(v)$ is $y'$.

- Configuration matrix $C_{v \rightarrow pa(v)}$, where element $C_{v \rightarrow pa(v)}(y', r)$ traces the label and rank $(y, r)$ of child $v$ that achieves $M_{v \rightarrow pa(v)}(y', r)$.

The processing of a node $v$ entails the following steps. First, the $K$-best lists of the children of the node stored in $M_{v' \rightarrow v}$ are merged in amortized $\Theta(K)$ time per child node. This is achieved by processing two child lists in tandem starting from the top of the lists and in each step picking the best pair of items to merge. This process results in the score matrix $P_v$ and the pointer matrix $C_v$.

Second, the $K$-best lists of $T_{v \rightarrow pa(v)}$ corresponding to all possible labels $y'$ of $pa(v)$ are formed. This is achieved by keeping the label of the head of the edge $v \rightarrow pa(v)$ fixed, and picking the best combination of labeling the tail of the edge and selecting a multilabel of $T_v$ consistent with that label. This process results in the matrices $M_{v \rightarrow pa(v)}$ and $C_{v \rightarrow pa(v)}$. Also this step can be performed in $\Theta(K)$ time.

The iteration ends when the root $v_{root}$ has updated its score $P_{v_{root}}$. Finally, the multilabels in form $\mathcal{Y}_{T_i, K}$ are traced using the pointers stored in $C_v$ and $C_{v \rightarrow pa(v)}$. The time complexity for a single tree is $\Theta(K\ell)$, and repeating the process for $n$ trees gives total time complexity of $\Theta(nK\ell)$.

## Master algorithm for training the model

The master algorithm (Algorithm 2) iterates over each training example until convergence. The processing of each training example proceeds by identifying the $K$ worst violators of each tree together with the threshold score $\theta_i = \theta_{x_i}$ (line 5), determining the worst ensemble violator from among them (line 6) and updating each tree by the worst ensemble violator (line 8). During the early stages of the algorithm, it is not essential to identify the worst violator. We therefore propose that initially $K = 2$, and the iterations continue until no violators are identified (line 7). We then increment $K$ and continue until the condition (line 10-12) given by Lemma 7 is satisfied so that we are assured of having converged to the global optimum.

**Algorithm 1** Algorithm to obtain top $K$ multilabels on a collection of spanning trees.

$FindKBest(\mathcal{T}, F_{E_{\mathcal{T}}}, K, \mathbf{y}_i)$

**Input:** Collection of rooted spanning trees $T_i = (E_i, V_i)$,
   edge labeling scores $F_{E_{\mathcal{T}}} = \{F_{T,v,v'}(y_v, y_{v'})\}$

**Output:** The best scoring multilabel $\mathbf{y}^*$, worst violator $\bar{\mathbf{y}}$, threshold $\theta_i$

1: **for** $T_i \in \mathcal{T}$ **do**
2:    Initialize $P_v, C_v, M_{v \to pa(v)}, C_{v \to pa(v)}, \forall v \in V_i$
3:    I = nodes indices in post-order of the tree $T_i$
4:    **for** $j = 1 : |I|$ **do**
5:       $v = v_{I(j)}$
6:       % Collect and merge $K$-best lists of children
7:       **if** $ch(v) \neq \emptyset$ **then**
8:          $P_v(y) = P_v(y) + \underset{r_v, v' \in ch(v)}{\textbf{kmax}} \left( \sum_{v' \in ch(v)} (M_{v' \to v}(y, r_v)) \right)$
9:          $C_v(y) = P_v(y) + \underset{r_v, v' \in ch(v)}{\textbf{argmax}} \left( \sum_{v' \in ch(v)} (M_{v' \to v}(y, r_v)) \right)$
10:      **end if**
11:      % Form the $K$-best list of $T_{v \to pa(v)}$
12:      $M_{v \to pa(v)}(y_{pa(v)}) = \underset{y, r}{\textbf{kmax}} \left( P_v(y, r) + F_{T, v \to pa(v)}(y_v, y_{pa(v)}) \right)$
13:      $C_{v \to pa(v)}(y_{pa(v)}) = \underset{u_v, r}{\textbf{argmax}} \left( P_v(u_v, r) + F_{T, v \to pa(v)}(u_v, y_{pa(v)}) \right)$
14:   **end for**
15:   Trace back with $C_v$ and $C_{v \to pa(v)}$ to get $\mathcal{Y}_{T_i, K}$.
16: **end for**
17: $\mathcal{Y}_{\mathcal{T}, K} = \bigcup_{T_i \in \mathcal{T}} \mathcal{Y}_{T_i, K}$
18: $\mathbf{y}^* = \underset{\mathbf{y} \in \mathcal{Y}_{\mathcal{T}, K}}{\operatorname{argmax}} \sum_{i=1}^{n} \sum_{\substack{(v, v') = \\ e \in E_i}} F_{T_i, v, v'}(y_v, y_{v'})$
19: $\bar{\mathbf{y}} = \underset{\mathbf{y} \in \mathcal{Y}_{\mathcal{T}, K} \backslash \mathbf{y}_i}{\operatorname{argmax}} \sum_{i=1}^{n} \sum_{\substack{(v, v') = \\ e \in E_i}} F_{T_i, v, v'}(y_v, y_{v'})$
20: $\theta_i = \sum_{i=1}^{n} \sum_{\substack{(v, v') = \\ e \in E_i}} F_{T_i, v, v'}(y_{T_i, K, v}, y_{T_i, k, v'})$

**Algorithm 2** Master algorithm.

**Input:** Training sample $\{(x_k, \mathbf{y}_k)\}_{k=1}^m$, collection of spanning trees $\mathcal{T}$, minimum violation $\gamma_0$
**Output:** Scoring function $F_{\mathcal{T}}$
1: $K_k = 2, \forall k \in \{1, \cdots, m\}; \mathbf{w}_{T_i} = 0, \forall T_i \in \mathcal{T}; converged = false$
2: **while** $not(converged)$ **do**
3:    $converged = true$
4:    **for** $k = \{1, \ldots, m\}$ **do**
5:       $S_{\mathcal{T}} = \{S_{T_i,e}(k, \mathbf{u}_e) | S_{T_i,e}(k, \mathbf{u}_e) = \langle \mathbf{w}_{T_i,e}, \phi_{T_i,e}(x_k, \mathbf{u}_e) \rangle, \forall (e \in E_i, T_i \in \mathcal{T}, \mathbf{u}_e \in \mathcal{Y}_v \times \mathcal{Y}_{v'}) \}$
6:       $[\mathbf{y}^*, \bar{\mathbf{y}}, \theta_i] = FindKBest(\mathcal{T}, S_{\mathcal{T}}, K_i, \mathbf{y}_i)$
7:       **if** $F_{\mathcal{T}}(x_i, \bar{\mathbf{y}}) - F_{\mathcal{T}}(x_i, \mathbf{y}_i) \geq \gamma_0$ **then**
8:          $\{\mathbf{w}_{T_i}\}_{T_i \in \mathcal{T}} = updateTrees(\{\mathbf{w}_{T_i}\}_{T_i \in \mathcal{T}}, x_i, \bar{\mathbf{y}})$
9:          $converged = false$
10:      **else**
11:        **if** $\theta_i > F_{\mathcal{T}}(x_i, \bar{\mathbf{y}})$ **then**
12:          $K_i = \min(2K_i, |\mathcal{Y}|)$
13:          $converged = false$
14:        **end if**
15:      **end if**
16:    **end for**
17: **end while**

## Derivation of the Marginal Dual

### Definition 6. Primal $L_2$-norm Random Tree Approximation

$$\min_{\mathbf{w}_{T_i}, \xi_k} \quad \frac{1}{2} \sum_{i=1}^n ||\mathbf{w}_{T_i}||_2^2 + C \sum_{k=1}^m \xi_k$$

$$\text{s.t.} \quad \sum_{i=1}^n \langle \mathbf{w}_{T_i}, \hat{\boldsymbol{\phi}}_{T_i}(x_k, \mathbf{y}_k) \rangle - \max_{\mathbf{y} \neq \mathbf{y}_k} \sum_{i=1}^n \langle \mathbf{w}_{T_i}, \hat{\boldsymbol{\phi}}_{T_i}(x_k, \mathbf{y}) \rangle \geq 1 - \xi_k$$

$$\xi_k \geq 0, \forall\, k \in \{1, \ldots, m\},$$

where $\{\mathbf{w}_{T_i} | T_i \in \mathcal{T}\}$ are the feature weights to be learned on each tree, $\xi_k$ is the margin slack allocated for each example $x_k$, and $C$ is the slack parameter that controls the amount of regularization in the model. This primal form has the interpretation of maximizing the joint margins from individual trees between (correct) training examples and all the other (incorrect) examples.

The Lagrangian of the primal form (Definition 6) is

$$\mathcal{L}(\mathbf{w}_{T_i}, \xi, \boldsymbol{\alpha}, \boldsymbol{\beta}) = \frac{1}{2} \sum_{i=1}^n ||\mathbf{w}_{T_i}||_2^2 + C \sum_{k=1}^m \xi_k - \sum_{k=1}^m \beta_k \xi_k$$
$$- \sum_{k=1}^m \sum_{\mathbf{y} \neq \mathbf{y}_k} \alpha_{k,\mathbf{y}} \left( \sum_{i=1}^n \langle \mathbf{w}_{T_i}, \Delta\hat{\boldsymbol{\phi}}_{T_i}(x_k, \mathbf{y}_k) \rangle - 1 + \xi_k \right),$$

where $\alpha_k$ and $\beta_k$ are Lagrangian multipliers that correspond to the constraints of the primal form, and $\Delta\hat{\boldsymbol{\phi}}_{T_i}(x_k, \mathbf{y}_k) = \hat{\boldsymbol{\phi}}_{T_i}(x_k, \mathbf{y}_k) - \hat{\boldsymbol{\phi}}_{T_i}(x_k, \mathbf{y})$. Note that given a training example-label pair $(x_k, \mathbf{y}_k)$ there are exponential number of $\alpha_{k,\mathbf{y}}$ one for each constraint defined by incorrect example-label pair $(x_k, \mathbf{y})$.

Setting the gradient of Lagrangian with respect to primal variables to zero, we obtain the following equalities:

$$\frac{\partial \mathcal{L}}{\partial \mathbf{w}_{T_i}} = \mathbf{w}_{T_i} - \sum_{k=1}^m \sum_{\mathbf{y} \neq \mathbf{y}_k} \alpha_{k,\mathbf{y}} \Delta\hat{\boldsymbol{\phi}}_{T_i}(x_k, \mathbf{y}_k) = 0,$$

$$\frac{\partial \mathcal{L}}{\partial \xi_k} = C - \sum_{\mathbf{y} \neq \mathbf{y}_k} \alpha_{k,\mathbf{y}} - \beta_k = 0,$$

which give the following dual optimization problem.

**Definition 10. Dual $L_2$-norm Random Tree Approximation**

$$\max_{\boldsymbol{\alpha}\geq 0} \quad \boldsymbol{\alpha}^{\mathsf{T}}\mathbf{1} - \frac{1}{2}\boldsymbol{\alpha}^{\mathsf{T}}\left(\sum_{i=1}^{n} K_{T_i}\right)\boldsymbol{\alpha}$$

$$\textit{s.t.} \quad \sum_{\mathbf{y}\neq\mathbf{y}_k} \alpha_{k,\mathbf{y}} \leq C, \forall\, k \in \{1,\ldots,m\},$$

where $\boldsymbol{\alpha} = (\alpha_{k,\mathbf{y}})_{k,\mathbf{y}}$ is the vector of dual variables. The joint kernel

$$
\begin{aligned}
K_{T_i}(x_k,\mathbf{y};x_{k'},\mathbf{y}') &= \langle \hat{\boldsymbol{\phi}}_{T_i}(x_k,\mathbf{y}_k) - \hat{\boldsymbol{\phi}}_{T_i}(x_k,\mathbf{y}), \hat{\boldsymbol{\phi}}_{T_i}(x_{k'},\mathbf{y}_{k'}) - \hat{\boldsymbol{\phi}}_{T_i}(x_{k'},\mathbf{y}')\rangle \\
&= \langle \varphi(x_k),\varphi(x_{k'})\rangle_{\varphi} \cdot \langle \psi_{T_i}(\mathbf{y}_k) - \psi_{T_i}(\mathbf{y}), \psi_{T_i}(\mathbf{y}_{k'}) - \psi_{T_i}(\mathbf{y}')\rangle_{\psi} \\
&= K^{\varphi}(x_k,x_{k'}) \cdot \left( K^{\psi}_{T_i}(\mathbf{y}_k,\mathbf{y}_{k'}) - K^{\psi}_{T_i}(\mathbf{y}_k,\mathbf{y}') - K^{\psi}_{T_i}(\mathbf{y},\mathbf{y}_{k'}) + K^{\psi}_{T_i}(\mathbf{y},\mathbf{y}')\right) \\
&= K^{\varphi}(x_k,x_{k'}) \cdot K^{\Delta\psi}_{T_i}(\mathbf{y}_k,\mathbf{y};\mathbf{y}_{k'},\mathbf{y}')
\end{aligned}
$$

is composed by input kernel $K^{\varphi}$ and output kernel $K^{\psi}_{T_i}$ defined by

$$K^{\varphi}(x_k,x_{k'}) = \langle \varphi(x_k),\varphi(x_{k'})\rangle_{\varphi}$$

$$K^{\Delta\psi}_{T_i}(\mathbf{y}_k,\mathbf{y};\mathbf{y}_{k'},\mathbf{y}') = K^{\psi}_{T_i}(\mathbf{y}_k,\mathbf{y}_{k'}) - K^{\psi}_{T_i}(\mathbf{y}_k,\mathbf{y}') - K^{\psi}_{T_i}(\mathbf{y}_{k'},\mathbf{y}) + K^{\psi}_{T_i}(\mathbf{y},\mathbf{y}').$$

To take advantage of the spanning tree structure in solving the problem, we further factorize the dual (Definition 10) according to the output structure [9, 16]. by defining a marginal dual variable $\mu$ as

$$\mu(k,e,\mathbf{u}_e) = \sum_{\mathbf{y}\neq\mathbf{y}_k} \mathbf{1}_{\{\psi(\mathbf{y})=\mathbf{u}_e\}}\alpha_{k,\mathbf{y}},$$

where $e = (j,j') \in E$ is an edge in the output graph and $\mathbf{u}_e \in \mathcal{Y} \times \mathcal{Y}'$ is a possible label of edge $e$. As each marginal dual variable $\mu(k,e,\mathbf{u}_e)$ is the sum of a collection of dual variables $\alpha_{k,\mathbf{y}}$ that has consistent label $(u_j,u_{j'}) = \mathbf{u}_e$, we have the following

$$\sum_{\mathbf{u}_e} \mu(k,e,\mathbf{u}_e) = \sum_{\mathbf{y}\neq\mathbf{y}_k} \alpha_{k,\mathbf{y}} \tag{12}$$

for an arbitrary edge $e$, independently of the structure of the trees.

The linear part of the objective (Definition 10) can be stated in term of $\boldsymbol{\mu}$ for an arbitrary collection of trees as

$$\boldsymbol{\alpha}^{\mathsf{T}}\mathbf{1} = \sum_{k=1}^{m} \sum_{\mathbf{y}\neq\mathbf{y}_k} \alpha_{k,\mathbf{y}} = \frac{1}{|E_{\mathcal{T}}|} \sum_{k=1}^{m} \sum_{e\in E_{\mathcal{T}}} \sum_{\mathbf{u}_e} \mu(k,e,\mathbf{u}_e) = \frac{1}{|E_{\mathcal{T}}|} \sum_{e,k,\mathbf{u}_e} \mu(k,e,\mathbf{u}_e),$$

where edge $e = (j,j') \in E_{\mathcal{T}}$ appearing in the collection of trees $\mathcal{T}$.

We observe that the label kernel of tree $T_i$, $K^{\psi}_{T_i}$, decomposes on the edges of the tree as

$$K^{\psi}_{T_i}(\mathbf{y},\mathbf{y}') = \langle \mathbf{y},\mathbf{y}'\rangle_{\psi} = \sum_{e\in E_i} \langle y_e,y'_e\rangle_{\psi} = \sum_{e\in E_i} K^{\psi,e}(y_e,y'_e).$$

Thus, the output kernel $K^{\Delta\psi}_{T_i}$ and the joint kernel $K_{T_i}$ also decompose

$$
\begin{aligned}
K^{\Delta\psi}_{T_i}(\mathbf{y}_k,\mathbf{y};\mathbf{y}_{k'},\mathbf{y}') &= \left( K^{\psi}_{T_i}(\mathbf{y}_k,\mathbf{y}_{k'}) - K^{\psi}_{T_i}(\mathbf{y}_k,\mathbf{y}') - K^{\psi}_{T_i}(\mathbf{y}_{k'},\mathbf{y}) + K^{\psi}_{T_i}(\mathbf{y},\mathbf{y}')\right) \\
&= \sum_{e\in E_i} \left( K^{\psi,e}_{T_i}(y_{ke},y_{k'e}) - K^{\psi,e}_{T_i}(y_{ke},y'_e) - K^{\psi,e}_{T_i}(y_e,y_{k'e}) + K^{\psi,e}_{T_i}(y_e,y'_e)\right) \\
&= \sum_{e\in E_i} K^{\Delta\psi,e}_{T_i}(y_{ke},y_e;y_{k'e},y'_e),
\end{aligned}
$$

$$
\begin{aligned}
K_{T_i}(x_k,\mathbf{y};x_{k'},\mathbf{y}') &= K^{\psi}(x_k,x_{k'}) \cdot K^{\Delta\psi}_{T_i}(\mathbf{y}_k,\mathbf{y};\mathbf{y}_{k'},\mathbf{y}') \\
&= K^{\psi}(x_k,x_{k'}) \cdot \sum_{e\in E_i} K^{\Delta\psi,e}(y_{ke},y_e;y_{k'e},y'_e) \\
&= \sum_{e\in E_i} K^{e}(x_k,y_e;x_{k'},y'_e).
\end{aligned}
$$

The sum of joint kernels of the trees can be expressed as

$$
\sum_{i=1}^{n} K_{T_i}(x_k, \mathbf{y}; x_{k'}, \mathbf{y}') = \sum_{i=1}^{n} \sum_{e \in E_i} K^e(x_k, y_e; x_{k'}, y'_e)
$$

$$
= \sum_{e \in E_{\mathcal{T}}} \sum_{\substack{T_i \in \mathcal{T}: \\ e \in E_i}} K^e(x_k, y_e; x_{k'}, y'_e)
$$

$$
= \sum_{e \in E_{\mathcal{T}}} N_{\mathcal{T}}(e) K^e(x_k, y_e; x_{k'}, y'_e)
$$

where $N_{\mathcal{T}}(e)$ denotes the number of occurrences of edge $e$ in the collection of trees $\mathcal{T}$.

Taking advantage of the above decomposition and of the Equation (12) the quadratic part of the objective (Definition 10) can be stated in term of $\boldsymbol{\mu}$ as

$$
-\frac{1}{2}\boldsymbol{\alpha}^{\mathsf{T}} \left( \sum_{i=1}^{n} K_{T_i} \right) \boldsymbol{\alpha}
$$

$$
= -\frac{1}{2}\boldsymbol{\alpha}^{\mathsf{T}} \left( \sum_{e \in E_{\mathcal{T}}} N_{\mathcal{T}}(e) K^e(x_k, \mathbf{y}; x_{k'}, \mathbf{y}') \right) \boldsymbol{\alpha}
$$

$$
= -\frac{1}{2} \sum_{k,k'=1}^{m} \sum_{e \in E_{\mathcal{T}}} N_{\mathcal{T}}(e) \sum_{\substack{\mathbf{y} \neq \mathbf{y}_k \\ \mathbf{y}' \neq \mathbf{y}_{k'}}} \alpha(k, \mathbf{y}) K^e(x_k, y_e; x_{k'}, y'_e) \alpha(k', \mathbf{y}')
$$

$$
= -\frac{1}{2} \sum_{k,k'=1}^{m} \sum_{e \in E_{\mathcal{T}}} N_{\mathcal{T}}(e) \sum_{\mathbf{u}_e, \mathbf{u}'_e} \sum_{\substack{\mathbf{y} \neq \mathbf{y}_k : y_e = \mathbf{u}_e \\ \mathbf{y}' \neq \mathbf{y}_{k'} : y'_e = \mathbf{u}'_e}} \alpha(k, \mathbf{y}) K^e(x_k, \mathbf{u}_e; x_{k'}, \mathbf{u}'_e) \alpha(k', \mathbf{y}')
$$

$$
= -\frac{1}{2} \sum_{k,k'=1}^{m} \sum_{e \in E_{\mathcal{T}}} \frac{N_{\mathcal{T}}(e)}{|E_{\mathcal{T}}|^2} \sum_{\mathbf{u}_e, \mathbf{u}'_e} \mu(k, e, \mathbf{u}_e) K^e(x_k, \mathbf{u}_e; x_{k'}, \mathbf{u}'_e) \mu(k', e, \mathbf{u}'_e)
$$

$$
= -\frac{1}{2} \sum_{\substack{e, k, \mathbf{u}_e, \\ k', \mathbf{u}'_e}} \mu(k, e, \mathbf{u}_e) K^e_{\mathcal{T}}(x_k, \mathbf{u}_e; x_{k'}, \mathbf{u}'_e) \mu(k', e, \mathbf{u}'_e),
$$

where $E_{\mathcal{T}}$ is the union of the sets of edges appearing in $\mathcal{T}$.

We then arrive at the following definition.

**Definition 9. Marginalized Dual $L_2$-norm Random Tree Approximation**

$$
\max_{\boldsymbol{\mu} \in \mathcal{M}^m} \quad \frac{1}{|E_{\mathcal{T}}|} \sum_{e, k, \mathbf{u}_e} \mu(k, e, \mathbf{u}_e) - \frac{1}{2} \sum_{\substack{e, k, \mathbf{u}_e, \\ k', \mathbf{u}'_e}} \mu(k, e, \mathbf{u}_e) K^e_{\mathcal{T}}(x_k, \mathbf{u}_e; x'_k, \mathbf{u}'_e) \mu(k', e, \mathbf{u}'_e),
$$

where $\mathcal{M}^m$ is marginal dual feasible set defined as (*c.f.*, [9])

$$
\mathcal{M}^m = \left\{ \boldsymbol{\mu} \,\middle|\, \mu(k, e, \mathbf{u}_e) = \sum_{\mathbf{y} \neq \mathbf{y}_k} \mathbf{1}_{\{\mathbf{y}_{ke} = \mathbf{u}_e\}} \boldsymbol{\alpha}(k, \mathbf{y}), \; \forall (k, e, \mathbf{u}_e) \right\}.
$$

The feasible set is composed of a Cartesian product of $m$ identical polytopes $\mathcal{M}^m = \mathcal{M} \times \cdots \times \mathcal{M}$, one for each training example. Furthermore, each $\boldsymbol{\mu} \in \mathcal{M}$ corresponds to some dual variable $\boldsymbol{\alpha}$ in the original dual feasible set $\mathcal{A} = \{\boldsymbol{\alpha} | \alpha(k, \mathbf{y}) \geq 0, \sum_{\mathbf{y} \neq \mathbf{y}_i} \alpha(k, \mathbf{y}) \leq C, \forall k\}$.

## Experimental Results

Table 2 provides the standard deviation results of the prediction performance results of Table 1 for each algorithm in terms of the microlabel and 0/1 error rates. Values are obtained by five fold cross-validation.

| DATASET | MICROLABEL LOSS (%) | | | | | 0/1 LOSS (%) | | | | |
|---|---|---|---|---|---|---|---|---|---|---|
| | SVM | MTL | MMCRF | MAM | RTA | SVM | MTL | MMCRF | MAM | RTA |
| EMOTIONS | 1.9 | 1.8 | 0.9 | 1.4 | 0.6 | 3.4 | 3.5 | 3.1 | 4.2 | 1.5 |
| YEAST | 0.7 | 0.5 | 0.6 | 0.5 | 0.6 | 2.8 | 1.0 | 1.5 | 0.4 | 1.2 |
| SCENE | 0.3 | 0.5 | 0.3 | 0.1 | 0.3 | 1.4 | 3.6 | 1.2 | 0.9 | 0.6 |
| ENRON | 0.2 | 0.2 | 0.2 | 0.2 | 0.2 | 0.3 | 0.4 | 2.8 | 2.3 | 0.9 |
| CAL500 | 0.3 | 0.3 | 0.3 | 0.2 | 0.4 | 0.0 | 0.0 | 0.0 | 0.0 | 0.0 |
| FINGERPRINT | 0.3 | 0.6 | 0.6 | 0.3 | 0.6 | 0.7 | 0.0 | 0.5 | 0.6 | 1.3 |
| NCI60 | 0.7 | 0.6 | 1.3 | 0.9 | 1.6 | 1.3 | 2.0 | 1.4 | 1.2 | 2.2 |
| MEDICAL | 0.0 | 0.1 | 0.1 | 0.1 | 0.2 | 2.1 | 2.3 | 3.3 | 2.5 | 3.6 |
| CIRCLE10 | 0.9 | 0.7 | 0.3 | 0.4 | 0.3 | 3.8 | 3.4 | 2.1 | 3.5 | 1.7 |
| CIRCLE50 | 0.5 | 0.5 | 0.3 | 0.3 | 0.6 | 2.0 | 3.3 | 4.5 | 5.5 | 2.2 |

Table 2: Standard deviation of prediction performance for each algorithm in terms of microlabel loss and 0/1 loss.