[Reviews · NeurIPS 2014]

Submitted by Assigned_Reviewer_11

The paper addresses the problem of learning structured output predictors using random tree estimators of the MAP problem.

The paper is clearly written and as far as I know original. The idea of approximating the MAP problem using random trees if very interesting, another strength of the method is its theoretical backing by the finite sample complexity, convergence bounds.

The main weakness in my opinion of the paper in my view is the experimental section. When comparing the Micro-label Loss, the differences between the algorithms seems quit minor. For instance if we compare to MAM alone, we get 5 losses 4 wins and one draws which is not very impressive. The results of the 0/1 loss are much more convincing, however their significance is not clear to me, as the algorithms where not trained directly on the 0/1 loss. In addition the practical importance of this measure is also not very convincing.
Summary: The work suggest a novel method to approximate structure prediction problems via random spanning trees. The work seem to have good theoretical founding, however its experimental part could be improved.

Submitted by Assigned_Reviewer_15

This paper provides the following method for approximate graphical model inference: first sample several spanning trees of the graphical model, only including the features corresponding to the edges in each tree; then build a scoring function that is the average of the scoring functions for each spanning tree. Note that the resulting function still could have arbitrarily large tree-width, even for an average of 2 trees; however, the authors provide an algorithm based on building a K-best list for each tree individually and then taking the best overall candidate from all of the lists. They provide a rigorous criterion under which K is large enough to admit exact inference, which is easy to check at runtime.

Quality: Overall I think this is a good paper; the authors make judicious use of theory, not proving things just for the sake of proving them but in order to answer the questions that would naturally occur to the reader of the paper.

Clarity: The paper is mostly clear but I think more care could be taken in reducing notational overhead, as well as explaining the significance of some of the theoretical results (in particular, I was unable to decipher Lemma 8).

Originality: The idea of using spanning trees for approximate inference is not new, but the particular approach that the authors take seems different from others. However, I would have liked to see a bit more explanation of the context and relation to other work (the authors do a good job of citing the relevant papers but not of explaining their relationship).

Significance: This paper has some interesting ideas that could potentially have broader impact. Lemma 7 (the stopping criterion for when inference is exact) is in my mind one of the strongest things going for this approach. The ideas in the paper seem flexible enough that they can potentially be extended and combined with other ideas.
Summary: Interesting paper with some potentially important ideas. The paper could be improved by providing more context and re-writing Lemma 8 to be more understandable.

Submitted by Assigned_Reviewer_30

***** Update after author feedback at the end *****
***************************************************

quality: 8 (out of 10)
clarity: 7
originality: 6
significance: 7

SUMMARY: The authors consider the use of random spanning trees to do structured prediction when the underlying graph structure on the labels is unknown. They motivate their approach by first showing that under some assumptions, the discriminant function over a fully connected graph on the labels can be expressed as the uniform expectation of the discriminant functions over random spanning trees. Through a sampling result, they then show that with high probability over samples of random spanning trees, there is a conical combination of these trees which achieve a substantial fraction of the margin of a predictor which uses the complete graph, and then prove a related risk bound for conical combination over random trees. This motivates to optimize the margin for conical combination of trees as predictors, and the author proposes a primal (and dual) formulation for this optimization problem (somewhat analog to the structured SVM), for which a standard dual subgradient method is proposed as in previous work. They then show that the maximizing joint label for the combination of trees (inference) can be done exactly (under an assumption that be checked at run-time) by looking through the K-best list for each spanning tree (the latter can be obtained by dynamic programming, as was already mentioned in Tsochantaridis et al. [JMLR 2005]). Experiments on standard multilabel datasets show a small improvement over alternative methods. Moreover, the authors investigate the proportion of times they can get an exact solution (according to the checkable condition they derived) as a function of K and the number of trees on the datasets. On some datasets, the exact solution is obtained a large fraction of time even for small K.

EVALUATION:

The authors claim in their introduction (line 051) "Our work is not based on any of these approaches, despite superficial resemblances to (for example) the trees in tree-based relaxations and the use of random trees in [5]." But I see this paper as a joint learning / joint inference formulation of [5] with a well-principled theoretical motivation (vs. the combination heuristics proposed in [5]). There is a consistent theoretical story behind the learning framework which is interesting (and I appreciate that all the proofs [except for theorem 5 -- see below] are provided. The paper is fairly clear to read and well-organized; on the other hand, there are several technical assumptions made for which it unclear how crucial they are for their results. They also seem more geared towards the multilabel setup than a general structured prediction problem. For example, they only consider the 0-1 loss in the theory (and not Hamming e.g.); and they assume that the weight vectors are not shared across the edges in the graph (unlike is often the case in typical structured prediction applications). They also assume that the features are normalized on each edge. How important are these assumptions for their results? I would like the authors to comment on this in their rebuttal.

I am not knowledgeable of the multilabel classification applications, so I cannot judge how significant are their empirical results. But the methodology seems sound; and it was interesting to observe that often the K-best lists were sufficient to obtain the global max.

Pros:
-A fairly complete theory motivating their learning objective.
- Combinations of random spanning trees seem like a reasonable approach to avoid working with the full graph when the graph is unknown given their explanations.
- Reasonable experimental comparison with somewhat practical algorithms.

Cons:
- Several technical assumptions are made along the way for which it is unclear how important they are to their results (or they are just there for simplicity).
- The authors don't really relate their work to the existing literature apart saying that the resemblances are "superficial".

== Other comments ==

- Line 204: they cite [8] for the risk bound -- but in [8] (e.g. Theorem 4.17), the risk bounds are only for binary labels as far as I could find. How do the authors adapt it to the exponential number of multilabels? Can they provide a more specific pointer?

=== Update after author feedback ===

Thank you for the clarifications.

My comment about the technical assumptions was to clarify which conditions are for simplicity of presentation or simplifying the statements, and which ones are essential to the current proofs. Indeed, I was referring to the normalization assumption for the feature map. In particular, a different normalization could be used in practice (even though "it can be performed for most phi"); and so could a different normalization be easily handled? Similarly, weight sharing and non 0-1 loss are important in structured prediction applications; how hard would it be to extend the results to these settings? The authors should clarify this in their final version (as well as the other aspects that they said they would clarify).
Summary: A theoretically well-founded paper motivating a somewhat practical algorithm for multilabel prediction with unknown graphs. A nice mix of theory, algorithms and experiments.
Author Feedback
Author rebuttal: Paper 564 : Author Feedback

First of all, we thank all three reviewers for their high-quality work.

To Assigned_Reviewer_11:

In our view, the 0/1 loss is useful for multilabel learning problems as it captures the fraction of multilabels that are perfectly predicted. Moreover, it is of no surprise that the differences in microlabel (Hamming) loss are less striking than the differences in 0/1 loss since our optimization problem (Definition 6) is written in terms of 0/1 loss - this is evident from the constraint that requires a margin of 1 (minus slack) from the worst violating multilabel (and, consequently, from all other incorrect multilabels). If we would rewrite the constraint in terms of the Hamming loss of the predicted multilabel, we would probably get a different behaviour with clearer differences in Hamming loss and, possibly, smaller differences in 0/1 loss.

To Assigned_Reviewer_15:

Lemma 8 provides a condition for which the highest scoring multilabel y_w(x) is present, with high probability, in one of the K-best lists. If it is present, Lemma 7 gives us a way to find it. But it is really Lemma 8 that gives us a condition for its presence! We think that the statement of Lemma 8 is clear (and correct) as long as we carefully read the text between Lemma 7 and 8. Lemma 8 basically states that if the scoring function F_T for each tree T is \sigma_T subgaussian (and we know that this is always the case for some \sigma_T > 0), then with very high probability over the random draws of n spanning trees, there exists a spanning tree T for which F_T(y_w(x)) is almost always larger than the score of F_T(y) of other multilabels y. This means that its rank K_T(y_w(x)) in tree T will be close to one. Following the statement of Lemma 8, we provide a specialization to the multiple binary classification case where show that, with high probability under the provided condition, there exists a tree T where K_T(y_w(x)) = 1.

To Assigned_Reviewer_30:

Theorem 4.17 of reference [8] applies to the class of pattern functions given by < w,\phi(x,y) > ; not just for the y < w,\phi(x) > class (as explicitly cited in the theorem). Hence here, y can be a member of an arbitrary large class. To see this, note that the empirical Rademacher complexity of the < w,\phi(x,y) > class is given by Theorem 4.12 of [8] where you just need to replace the input kernel k(x,x’) with < \phi(x,y),\phi(x’,y’) >. As explained just before Theorem 5, w consists of the concatenation of n unit L2 norms weight vectors w_{T_i}; each weighted by q_i. Hence, w has a unit L2 norm. Also, \phi(x,y) is obtained by concatenating each unit L2 norm single-tree feature vector multiplied by 1/\sqrt{n}. So that the trace of the kernel matrix <\phi(xi,yi),\phi(xj,yj)> is equal to the number m of training examples. This explains the 2/(\gamma \sqrt{m}) term in the bound of Theorem 5. The factor of 2 (instead of 4 in Theorem 4.17 of [8]) is due to a tighter version of the Lipschitz composition theorem (Theorem 4.15 iv of [8]) which is stated in Lemma 1.1 of Lecture 17 of Kakade’s Learning Theory course. We will add this to the supplementary material. Thanks for pointing this out.

Several technical assumptions? The basic assumption is that \phi(x,y) on the complete graph G has unit L2 norm and that for each edge (i,j) of G, \phi_{i,j}(x,y_i, y_j) has the same norm. This is just a convenient normalization which can be performed for most \phi_{i,j}(). The (very few) remaining assumptions are provided in the lemmas and theorems. Note that when we state (in Lem2 and Thm4) that w achieves a margin of \Gamma(w,x,y) on G for all (x,y), this is not an assumption as this margin function is arbitrary. Yes, we have provided results for the 0-1 loss via the ramp loss (i.e., the clipped hinge loss). We have not provided results for the Hamming or other microlabel losses. We view these as extensions rather than assumptions.

For multilabel problems, it is not clear that weight sharing is beneficial as the microlabels generally need to be assumed to be heterogeneous (corresponding to arbitrary properties of the target with, potentially, distinctly different marginal distributions). This makes the problems different from image segmentation or sequence annotation where one can assume some level of homogeneity among the components of the target object. It is of course possible that some multilabel problems would benefit from weight sharing. However, we will leave this for future investigation.

Comparing to previous methods, to our knowledge, this is the first time that approximate inference with spanning trees of a general graph has been fully intertwined with learning a model for labelling the graph. Conceptually closest to the present method is the MAM method by Su and Rousu (2013) which optimizes structured prediction models on the spanning trees independently, and then composes the ensemble prediction by different aggregation approaches. The tree-based reparametrization framework of Wainwright, Jaakkola and Wilsky (2005) was a source of initial inspiration to us. However, their approach is a pure inference method where the parameters are assumed to be learnt already. In our approach, each update of the model parameters involves inference over the spanning trees. We will clarify this in the final version.

We agree with the reviewer that the main difference between [5] is to bring in the joint learning/inference framework and to provide a robust learning theory to back up the algorithms. In our view, this is a major step forward.